# UMEM: Unified Memory Extraction and Management Framework for Generalizable Memory

Yongshi Ye [1 2]   Hui Jiang [3]   Feihu Jiang [3]   Tian Lan [3]   Yichao Du [3]   Biao Fu [2]   Xiaodong Shi [1 2]   Qianghuai Jia [3]   Longyue Wang [3]   Weihua Luo [3]

## Abstract

Self-evolving memory serves as the trainable parameters for Large Language Models (LLMs)-based agents, where extraction (distilling insights from experience) and management (updating the memory bank) must be tightly coordinated. Existing methods predominantly optimize memory management while treating memory extraction as a static process, resulting in poor generalization, where agents accumulate instance-specific noise rather than robust memories. To address this, we propose **U**nified **M**emory **E**xtraction and **M**anagement (UMEM), a self-evolving agent framework that jointly optimizes a LLM to simultaneously extract and manage memories. To mitigate overfitting to specific instances, we introduce Semantic Neighborhood Modeling and optimize the model with a neighborhood-level marginal utility reward via GRPO. This approach ensures memory generalizability by evaluating memory utility across clusters of semantically related queries. Extensive experiments across five benchmarks demonstrate that UMEM significantly outperforms highly competitive baselines, achieving up to 11.49 points improvement in multi-turn interactive tasks. Furthermore, UMEM maintains a stable improvement trend during continuous evolution.

## 1. Introduction

Self-evolution is a fundamental capability for agents operating in dynamic, open-ended environments (Zhang et al., 2026). While Large Language Models (LLMs) serve as powerful backbones for agents, their parameters typically remain frozen after deployment, limiting their ability to learn from continuous interactions. To overcome this limitation, long-term memory serves as trainable parameters of agents that can be updated from online experience (Cai et al., 2025b;a; Ouyang et al., 2025; Wei et al., 2025).

Conceptually, self-evolving agents mirror the neural network optimization (Rumelhart et al., 1986; Cai et al., 2025b; Ouyang et al., 2025): (1) Forward Pass: the frozen agent executes a task given retrieved memories from memory bank; and (2) Backward Optimization: a memory optimizer extracts memories from the execution trajectory and consolidates them into the memory bank (Xu et al., 2025; Yan et al., 2025). Therefore, the bottleneck of the self-evolving agent lies in the capability of this memory optimizer.

While numerous works have improved the memory optimizer, they predominantly focus on memory management, treating extraction as a static process via prompting off-the-shelf LLMs (Wu et al., 2025; Yan et al., 2025; Fang et al., 2025), without optimizing for explicit generalization. Consequently, self-evolving agents suffer from two critical problems: (1) **Accumulation of Instance-Specific Noise**: As shown in Figure 1, static memory extraction blindly retains instance-specific details rather than generalizable principles (Qin et al., 2024), causing progressive memory pollution and poor generalization; (2) **Management Misalignment**: The extracted memories are often inconsistent with the corresponding management policy, rendering even an optimal management policy ineffective. Therefore, even a well-optimized management policy cannot compensate for low-quality memories, undermining both task performance and cross-task generalization of the self-evolving agents.

To bridge this gap, we propose **Unified Memory Extraction and Management (UMEM)**, a self-evolving agent framework that jointly optimizes the memory extraction and management capability of memory optimizer. Structurally, UMEM consists of three primary components: a frozen Agent Executor (inference engine), a Memory Bank (the external parameters of self-evolving agents), and a learned memory optimizer (Mem-Optimizer). The Mem-

[1]Institute of Artificial Intelligence, Xiamen University [2]Key Laboratory of Digital Protection and Intelligent Processing of Intangible Cultural Heritage of Fujian and Taiwan (Xiamen University), Ministry of Culture and Tourism [3]Alibaba Group. Correspondence to: Xiaodong Shi <mandel@xmu.edu.cn>, Longyue Wang <wanglongyue.wly@alibabainc.com>.

*Proceedings of the 43rd International Conference on Machine Learning*, Seoul, South Korea. PMLR 306, 2026. Copyright 2026 by the author(s).

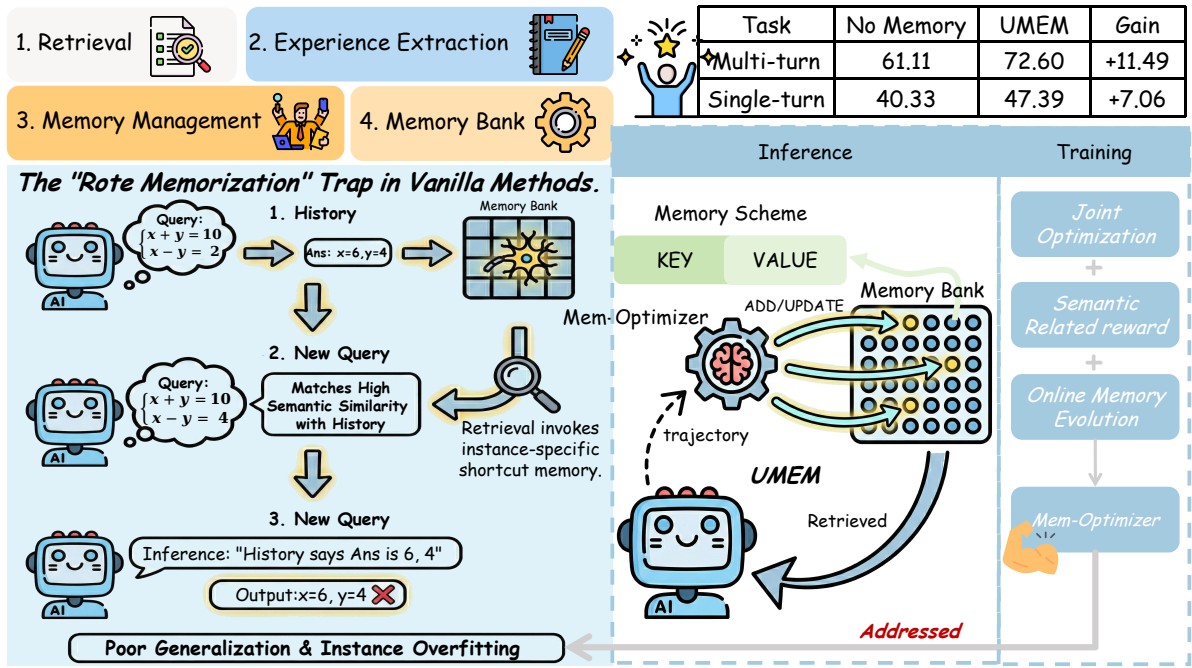

*Figure 1.* Comparison between the conventional memory pipeline and our proposed UMEM framework. **Left**: Vanilla methods suffer from the "Rote Memorization" trap, overfitting to instance-specific noise. **Right**: UMEM utilizes a learnable Mem-Optimizer to jointly optimize extraction and management. This distills generalizable principles, ensuring robust performance and avoiding noise accumulation.

Optimizer stands as the core of our proposed UMEM framework, designed to evolve the memory bank by extracting reusable memories from executor's trajectory. Crucially, to address the instance-specific noise, we introduce the Semantic Neighborhood Modeling, which constructs clusters of semantically related queries to simulate cross-task variations, and design a Marginal Utility Reward to guide the optimization process. By maximizing this reward via Group Relative Policy Optimization (GRPO), Mem-Optimizer performs end-to-end joint optimization. This guarantees that extracted memories are not only generalizable but also intrinsically aligned with the management policy. Besides, we implement Online Memory Evolution, where the memory bank is dynamically updated with optimal rollouts during training, forcing the agent to learn how to utilize a continuously refining memory system. Ultimately, the trained Mem-Optimizer significantly enhancing the cross-task generalization capability of agents.

Extensive experiments across five benchmarks demonstrate that UMEM significantly outperforms highly competitive baselines like MemRL and $Mem^p$ on single-turn reasoning tasks. Notably, ablation studies demonstrate that optimizing memory management in isolation leads to significant performance degradation, empirically validating the necessity of jointly optimizing memory extraction and management. Further analysis confirms that Semantic Neighborhood Modeling and the Marginal Utility Reward Function effectively empower the Mem-Optimizer to distill generalizable memo-

ries from individual experiences, rather than merely memorizing instance-specific shortcuts. Finally, results of test-time scaling evolution prove that UMEM enables agents to achieve robust and stable self-evolution, maintaining a consistent performance gain and widening the performance gap compared to baselines as interactions proceed. These designs ensure our proposed UMEM could effectively transform interaction experience into helpful insights, paving the way for truly self-evolving agents.

**Conflict of Interest Disclosure.** Several authors are employed by Alibaba Group, which develops the Qwen model family evaluated in this paper.

## 2. Related Work

**From Parametric Memory to Non-Parametric Memory.** Research on memory-augmented language models has spanned from early architectural mechanisms (Weston et al., 2015; Borgeaud et al., 2022; Liu et al., 2024) to recent scalable lookup frameworks (Lan et al., 2023; Cheng et al., 2026). However, these approaches require costly fine-tuning. Recent work has converged on a non-parametric paradigm: treating the external memory bank as the agent's evolvable parameters (Wei et al., 2025; Cai et al., 2025b;a).

**Self-Evolving Memory without Optimization.** The effectiveness of non-parametric evolution hinges on *how* experiences are represented. Initial attempts, such as

Synapse (Zheng et al., 2024), retrieved raw historical trajectories. However, this approach suffers from severe noise and context window inefficiencies. To distill clearer signals, subsequent works introduced structured abstraction. For example, $Mem^p$ (Fang et al., 2025) converts trajectories into executable programs. ReasoningBank (Ouyang et al., 2025) summarizes success and failure trajectories into reusable memory entries. However, the memory extraction and management policy of these methods mainly rely on prompting LLMs or hand-crafted rules, preventing the further improvement of extraction and management capability.

**Self-Evolving Memory with Optimization.** Recent research integrates optimization, such as Reinforcement Learning (RL), into self-evolving agents, with two streams: (1) **Optimizing Working Memory or Short-term Memory**: DeepAgent (Li et al., 2025), MemAgent (Xu et al., 2025) and Mem-$\alpha$ (Wang et al., 2025) employ RL to manage working memory or short-term memory (Jiang et al., 2025). While effective for handling long-context inputs, they do not construct an evolvable memory bank, which falls outside the scope of our comparison; (2) **Optimizing Long-term Memory**: This stream aims to enhance agents' memory selection and management capabilities, exemplified by Memory-R1 (Yan et al., 2025) and MemRL (Zhang et al., 2026). EvolveR (Wu et al., 2025) further introduces explicit rules to manage distilled experience principles. Existing works exhibit a critical limitation: they predominantly optimize memory selection and management while treating memory extraction as a static process (Yan et al., 2025). Furthermore, they lack explicit mechanisms to model generalization across future queries, often resulting in the accumulation of low-quality, instance-specific noise. In contrast, we propose the UMEM framework to jointly optimize memory extraction and management policy, ensuring that evolved memories are generalizable and aligned with future reuse.

## 3. Task Formulation of Self-Evolving Agents

Self-evolving agent can be treated as a parametric system where the executor $\mathcal{E}$ (parameters $\Theta_0$) are frozen, and the external memory bank $\mathcal{B}$ serves as the evolvable, non-differentiable parameters, consisting of a set of key–value pairs $\mathcal{B} = \{(k_i, v_i)\}_{i=1}^{|\mathcal{B}|}$. Here, each key $k_i$ is the embedding of the source query encoded by BGE-M3 (Chen et al., 2024), and each value $v_i$ stores the associated memory content extracted from the corresponding executor trajectory. In our proposed UMEM, the self-evolving process of agents is conceptualized as analogous to a network optimization process, comprising a forward pass for inference and a backward optimization for memory evolution.

**Feedforward Pass (Memory-Augmented Execution).** At time $t$, given a query $q$, the agent encodes $q$ with the same BGE-M3 encoder and retrieves the Top-$K$ relevant memory entries $\mathcal{B}_t^{topk} \in \mathcal{B}_t$ according to cosine similarity between the query embedding and memory keys. Then, the frozen executor $\mathcal{E}$ performs inference conditioned on this context to generate a complete trajectory $\tau_q$ and prediction $\hat{y}_q$:

$$\tau_q, \hat{y}_q \leftarrow \mathcal{E}(q, \mathcal{B}_t^{topk}; \Theta_0)$$

Here, since $\Theta_0$ is fixed, the system's performance is strictly bounded by the quality of the retrieved memory $\mathcal{B}_t^{topk}$.

**Backward Pass (Memory Bank Update).** The key to the self-evolving memory is to optimize the memory bank $\mathcal{B}$. Since $\Theta_0$ is fixed, the system's performance is strictly bounded by the quality of the memory bank $\mathcal{B}_t$. Analogous to a backward optimization process, a Memory Optimizer model (Mem-Optimizer), parameterized by $\phi$, extracts memory entries (distills insights) from the trajectory $\tau_q$, and samples a pre-defined memory management operation $opt_q \in \{\texttt{ADD}, \texttt{UPDATE}\}$:

$$a_q = (\Delta_q, opt_q) \sim \pi_\phi(\cdot \mid q, \tau_q, \hat{y}_q)$$

where $\Delta_q$ is the extracted memory and $a_q$ represents the action to the memory bank; $\texttt{ADD}$ inserts $(e(q), \Delta_q)$ as a new entry, whereas $\texttt{UPDATE}$ revises the value of a selected retrieved entry while keeping its key fixed.

The memory bank evolves after applying the action: $\mathcal{B}_{t+1} \leftarrow \text{Apply}(\mathcal{B}_t, a_q)$. Note that while we formulate the input as the current trajectory $\tau_q$, this representation is generic; it can easily extend to extracting insights from pairs of successful or failed trajectories (Ouyang et al., 2025). In conclusion, identifying the Mem-Optimizer ($\pi_\phi$) as the core bottleneck (Zhang et al., 2026; Fang et al., 2025; Cai et al., 2025b), we propose the UMEM framework to jointly optimize its extraction and management policies.

## 4. Method

This section presents the UMEM framework. To improve generalization, UMEM combines Semantic Neighborhood Modeling, which constructs query clusters to reduce overfitting (Section 4.1), with a Marginal Utility Reward optimized by GRPO to enforce cross-task utility (Section 4.2).

### 4.1. Semantic Neighborhood Modeling

A critical risk in memory evolution is *overfitting*: an extracted insight may perfectly resolve the current query but fail to generalize to related queries due to instance-specific noise or shortcuts (Qin et al., 2024). To mitigate this, we introduce Semantic Neighborhood Modeling. Our core insight is to treat the local cluster of similar queries as a proxy to approximate cross-task variations. Specifically, we first project all queries into a shared semantic space using a pre-trained encoder (e.g., BGE-M3 (Chen et al., 2024)). For a given

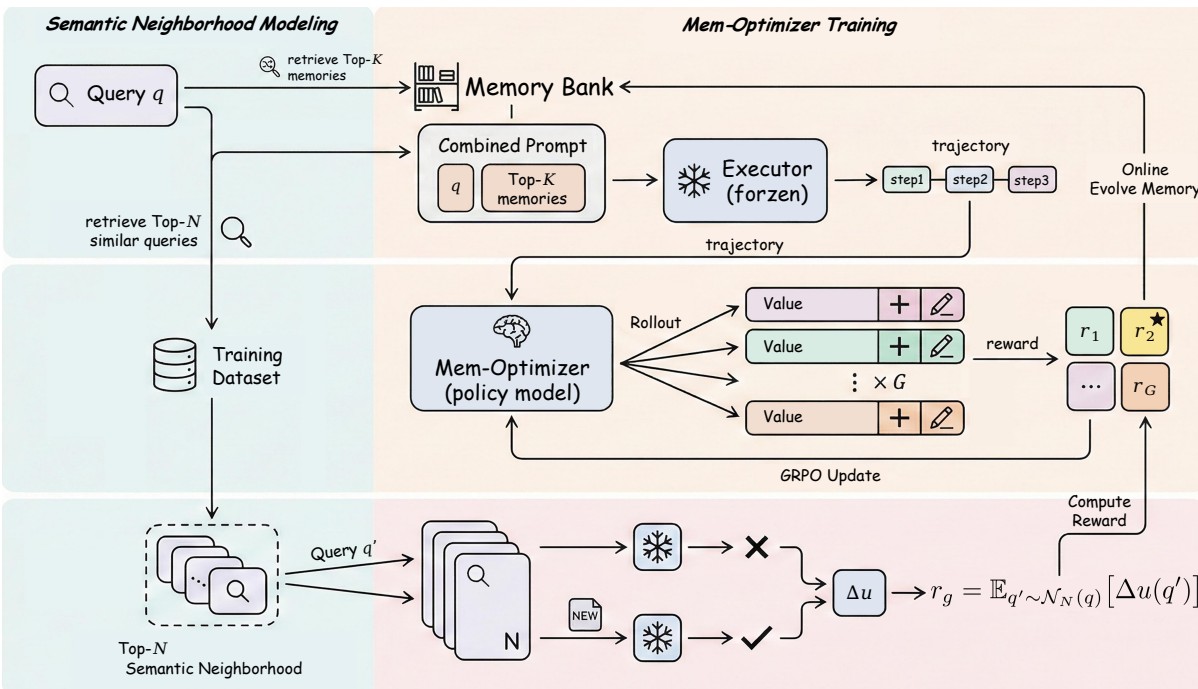

*Figure 2.* Overview of UMEM. **Left**: Semantic Neighborhood Modeling retrieves related queries to simulate cross-task variations. **Right**: The Mem-Optimizer distills trajectories from the frozen Executor into memory updates, which are optimized via GRPO. The process is guided by a Marginal Utility Reward that measures performance gains across the entire neighborhood to ensure generalization.

source query $q$, we construct its semantic neighborhood $\mathcal{N}_N(q)$ by retrieving the Top-$N$ nearest neighbors from the corpus $\mathcal{D}$ based on cosine similarity. During training, we evaluate candidate memory updates not on the current $q$, but over the entire neighborhood $\mathcal{N}_N(q)$. This mechanism forces the Mem-Optimizer to discard instance-specific details and extract generalizable insights.

### 4.2. Mem-Optimizer Training via GRPO

Mem-Optimizer training has five stages: (1) Memory-Augmented Execution; (2) Policy Rollout; (3) Marginal Utility Reward; (4) GRPO Optimization; and (5) Online Memory Evolution. Appendix E provides the procedure.

**Memory-Augmented Execution.** As described in Section 3, for each query in training dataset $q \in \mathcal{Q}$ at training step $t$, we retrieve the Top-$K$ relevant memory entries $\mathcal{B}_t^{topk}$ from the current memory bank. The frozen executor then generates a trajectory $\tau_q$ and prediction $\hat{y}_q$.

**Mem-Optimizer Policy Rollout.** As shown in the middle of the right panel of Figure 2, the Mem-Optimizer distills the $\tau_q$ into structured memory action $(\Delta_q, opt_q)$ (values). Adopting the GRPO algorithm (Shao et al., 2024), we sample a group of $G$ memory update actions $\{a_q^{(g)}\}_{g=1}^G$:

$$\{a_q^{(g)}\}_{g=1}^G \sim \pi_\phi(\cdot \mid q, \tau_q, \mathcal{B}_t^{topk}) \tag{1}$$

**Marginal Utility Reward.** To evaluate the quality of the generated memory update actions $\{a_q^{(g)}\}_{g=1}^G$, we strictly prohibit overfitting to the single source query $q$. Instead, we validate the memory update against the Semantic Neighborhood $\mathcal{N}_N(q)$. For each neighbor query $q' \in \mathcal{N}_N(q)$, we compute the per-neighbor utility $\Delta u(q')$ by comparing two execution states: a reference execution without $a_q^{(g)}$ and a memory-augmented execution where $a_q^{(g)}$ is used. The marginal utility contains two terms: (1) **Success Gain** ($\mathcal{G}_{\text{succ}}$): It quantifies the correction of execution failures:

$$\mathcal{G}_{\text{succ}}(q') = c(\tau_{q'}^{\text{mem}}) - c(\tau_{q'}^{\text{ref}}) \tag{2}$$

where $c(\tau_{q'}^{\text{mem}}), c(\tau_{q'}^{\text{ref}}) \in \{0, 1\}$ denote the correctness of the augmented and reference execution trajectories, respectively. A positive $\mathcal{G}_{\text{succ}}$ indicates that the memory successfully fixed a previously incorrect query, while a negative value penalizes memory that introduces errors into originally correct reasoning; (2) **Efficiency Regularization ($\mathcal{R}_{\text{eff}}$):** Beyond correctness, high-quality memory should facilitate more efficient inference, pruning redundant and wrong reasoning steps (Ahmed et al., 2025; Didolkar et al., 2025). To encourage concise reasoning, we introduce an Efficiency Regularization term that rewards token reduction, but **only** when correctness is preserved:

$$\mathcal{R}_{\text{eff}}(q') = (c_{\text{mem}} \cdot c_{\text{ref}}) \cdot \left(1 - \frac{\ell_{\text{mem}}^{(g)}}{\ell_{\text{ref}}}\right) \tag{3}$$

Here, $\ell$ represents the length of the generated trajectory. The gating term $(c_{\text{mem}} \cdot c_{\text{ref}})$ ensures that we do not reward brevity if it comes at the cost of accuracy (e.g., generating a short but wrong answer). The marginal utility reward is then defined as the sum of these two reward scores:

$$\Delta u(q') = \mathcal{G}_{\text{succ}}(q') + \mathcal{R}_{\text{eff}}(q'). \tag{4}$$

The final marginal utility reward for a candidate memory is the average marginal utility over the neighborhood,

$$r_g = \mathbb{E}_{q' \sim \mathcal{N}_N(q)} \big[ \Delta u(q') \big],$$

which favors memory updates that both correct errors of semantically related queries.

**Optimization via GRPO.** Finally, we train $\pi_\phi$ to maximize a joint objective $r_{\text{final}} = r_{\text{fmt}} + r_g$, where $r_{\text{fmt}} \in \{0, 1\}$ is a binary format reward that validates if the output format of extracted memories and management operations strictly adheres to the XML schema defined in Appendix C.

**Online Memory Evolution.** After GRPO optimization of one query $q$, we identify the memory update action $a_q^{(g)}$ with the highest marginal utility reward and immediately apply it to the memory bank: $\mathcal{B}_{t+1} \leftarrow \text{Apply}(\mathcal{B}_t, a_q)$. This ensures that the memory bank is dynamically refined throughout the training process, forcing the agent to learn how to utilize and manage an evolving rather than static memory.

## 5. Experiments

### 5.1. Setup

**Datasets.** We derive our training data from the MMLU dataset (Hendrycks et al., 2021). Specifically, we randomly sample $\sim$2,000 queries. For each query $q$, a semantic neighborhood cluster $\mathcal{N}_N(q)$ is constructed by retrieving the Top-$N$ ($N = 3$) most similar samples within the training set with a similarity threshold of 0.6.

**Backbone.** We employ Llama-3.2-1B-Instruct and Qwen3-4B-Instruct as the Mem-Optimizer policy $\pi_\phi$. During training (details in Appendix B), Qwen3-8B serves as the frozen executor $\mathcal{E}$ to generate trajectories. To evaluate cross-model portability, we deploy the Mem-Optimizer to curate memory for diverse unseen executors, including GPT-5.1, Qwen3-8B, and Gemini-2.5-Flash. This setup assesses whether UMEM distills architectural-agnostic insights that generalize to heterogeneous and stronger models. We provide an itemized training-cost analysis in Appendix G; reward evaluation is the dominant bottleneck because it requires executor-based evaluation over semantic neighborhoods.

**Baselines.** We evaluate UMEM against several representative paradigms: (1) No Memory, which assesses the frozen backbone LLM without external memory; (2) No Train, a non-learning ablation using identical prompt templates without policy training; (3) Self-RAG (Asai et al., 2024), which filters retrieved context via inference-time self-critique; (4) $Mem^p$ (Fang et al., 2025), a decoupled Build-Retrieve-Update pipeline that distills trajectories into procedural memories; (5) ReMem (Wei et al., 2025), a baseline focusing on memory management that maintains trajectory-level memory via discrete operations interleaved with reasoning steps; (6) ReasoningBank (Ouyang et al., 2025), which distills trajectories into reasoning memories without an explicit memory-management module; (7) MemRL (Zhang et al., 2026), which learns utility-guided memory selection and management via runtime RL; and (8) EvolveR (Wu et al., 2025), whose full policy-evolution pipeline does not fit our frozen-executor setting, so we adopt its explicit rules for managing distilled experience. Unlike these methods, UMEM uniquely targets the joint optimization and granularity alignment of memory extraction and management.

**Benchmark.** We evaluate UMEM on five benchmarks covering single-turn reasoning and multi-turn embodied interaction. For single-turn tasks, we select AIME (AIME24 and 25) (Hugging Face H4, 2024; OpenCompass, 2025) and GPQA-Diamond (Rein et al., 2023) for mathematical and scientific reasoning, alongside the multiple-choice subset of HLE (Phan et al., 2025) for multidisciplinary reasoning. We also include the first 100 HotpotQA examples (Yang et al., 2018) to evaluate strategy reuse in multi-hop question answering. For these single-turn benchmarks, performance is reported using Exact Match (EM) accuracy. For multi-turn embodied settings, we adopt ALFWorld (Shridhar et al., 2021), which requires long-horizon planning and state-dependent decision-making; we report Cumulative Success Rate (CSR) and Progress Rate following prior benchmark/-metric practice (Wu et al., 2024; Wei et al., 2025).

**Evaluation Protocol.** We adopt a streaming protocol to assess the agent's continuous self-evolution. Unlike static benchmarks, tasks are processed as a sequential stream. This zero-reset setup ensures that experiences distilled from trajectory are immediately integrated into memory bank to facilitate the reasoning of all subsequent queries.

### 5.2. Main Results

As illustrated in Table 1, UMEM consistently outperforms all baseline methods, including state-of-the-art memory management systems like MemRL and $Mem^p$, across the vast majority of benchmarks. Notably, our framework achieves significant performance leaps in complex reasoning tasks (e.g., AIME and GPQA_Diamond) and embodied environments like ALFWorld, where UMEM-Qwen3-4B attains a Success Rate of 83.09% when paired with GPT-5.1.

A key observation is that the effectiveness of UMEM is positively correlated with the strength of the frozen executor;

*Table 1.* Main Results. We evaluate UMEM using three distinct frozen executors: Qwen3-8B-Thinking, GPT-5.1, and Gemini-2.5-Flash. Scores are reported as Avg@3, computed as mean ± standard error over three runs with randomized benchmark orders.

| Models | Parameters | AIME | GPQA | HLE | HotpotQA | ALFWorld | | Average |
| --- | --- | --- | --- | --- | --- | --- | --- | --- |
| | | | | | | SR | PR | |
| *Frozen Executor: Qwen3-8B-Thinking* | | | | | | | | |
| No Memory | - | 51.67 | 52.53 | 7.51 | 62.00 | 41.04 | 68.91 | 47.28 |
| Self-RAG | 8B | $31.11_{\pm0.45}$ | $43.60_{\pm0.27}$ | $8.38_{\pm0.15}$ | $24.67_{\pm0.27}$ | $30.85_{\pm0.20}$ | $55.39_{\pm0.53}$ | $32.33_{\pm0.12}$ |
| ReMem | 8B | $\mathbf{62.22}_{\pm0.45}$ | $52.86_{\pm1.46}$ | $5.53_{\pm0.52}$ | $17.33_{\pm0.98}$ | $48.51_{\pm1.27}$ | $68.32_{\pm0.97}$ | $42.46_{\pm0.39}$ |
| $Mem^p$ | 8B | $47.22_{\pm1.20}$ | $49.66_{\pm0.60}$ | $\mathbf{9.98}_{\pm0.52}$ | $61.67_{\pm0.27}$ | $45.03_{\pm0.54}$ | $68.57_{\pm1.41}$ | $47.02_{\pm0.56}$ |
| MemRL | 8B | $48.89_{\pm1.20}$ | $50.34_{\pm0.37}$ | $9.09_{\pm0.25}$ | $61.67_{\pm0.27}$ | $50.25_{\pm0.20}$ | $58.17_{\pm0.79}$ | $46.40_{\pm0.31}$ |
| ReasoningBank | 8B | $56.67_{\pm0.78}$ | $49.50_{\pm1.04}$ | $7.31_{\pm0.15}$ | $59.33_{\pm1.09}$ | $41.05_{\pm0.93}$ | $50.58_{\pm1.38}$ | $44.07_{\pm0.48}$ |
| EvolveR | 8B | $23.89_{\pm0.45}$ | $52.53_{\pm1.26}$ | $7.31_{\pm0.15}$ | $61.00_{\pm0.00}$ | $34.82_{\pm0.20}$ | $44.61_{\pm0.59}$ | $37.36_{\pm0.18}$ |
| Llama-3.2-1B-Instruct | 1B | $52.22_{\pm0.45}$ | $52.36_{\pm0.14}$ | $6.77_{\pm0.52}$ | $59.33_{\pm0.27}$ | $45.77_{\pm0.73}$ | $70.94_{\pm1.38}$ | $47.90_{\pm0.20}$ |
| **UMEM-Llama-3.2-1B (Ours)** | 1B | $57.78_{\pm1.20}$ | $54.55_{\pm0.24}$ | $8.74_{\pm0.89}$ | $60.33_{\pm0.27}$ | $45.27_{\pm0.20}$ | $71.10_{\pm0.42}$ | $49.63_{\pm0.08}$ |
| Qwen3-4B-Instruct | 4B | $55.56_{\pm0.45}$ | $53.37_{\pm0.60}$ | $6.60_{\pm0.14}$ | $60.33_{\pm0.27}$ | $40.55_{\pm0.20}$ | $67.46_{\pm0.88}$ | $47.31_{\pm0.20}$ |
| **UMEM-Qwen3-4B (Ours)** | 4B | $58.89_{\pm0.45}$ | $53.54_{\pm0.24}$ | $8.38_{\pm0.15}$ | $63.00_{\pm0.00}$ | $50.25_{\pm0.41}$ | $74.17_{\pm0.43}$ | $51.37_{\pm0.10}$ |
| Qwen3-14B | 14B | $57.22_{\pm0.45}$ | $53.71_{\pm0.14}$ | $7.84_{\pm0.14}$ | $61.67_{\pm0.27}$ | $47.51_{\pm0.89}$ | $70.69_{\pm0.80}$ | $49.77_{\pm0.45}$ |
| **UMEM-Qwen3-14B (Ours)** | 14B | $60.56_{\pm0.45}$ | $\mathbf{55.05}_{\pm0.29}$ | $8.74_{\pm0.52}$ | $\mathbf{63.67}_{\pm0.27}$ | $\mathbf{52.24}_{\pm0.93}$ | $\mathbf{74.55}_{\pm0.42}$ | $\mathbf{52.47}_{\pm0.48}$ |
| *Frozen Executor: GPT-5.1* | | | | | | | | |
| No Memory | - | 40.00 | 57.57 | 6.95 | 39.00 | 61.94 | 66.67 | 45.36 |
| Self-RAG | API | $48.33_{\pm0.78}$ | $57.24_{\pm0.27}$ | $7.67_{\pm0.14}$ | $54.33_{\pm0.27}$ | $67.41_{\pm1.77}$ | $79.15_{\pm1.88}$ | $52.36_{\pm0.70}$ |
| ReMem | API | $33.33_{\pm1.57}$ | $64.31_{\pm0.83}$ | $8.92_{\pm0.77}$ | $44.00_{\pm0.47}$ | $68.41_{\pm2.12}$ | $76.82_{\pm1.45}$ | $49.30_{\pm0.11}$ |
| $Mem^p$ | API | $43.33_{\pm0.78}$ | $62.29_{\pm0.83}$ | $8.56_{\pm0.67}$ | $52.00_{\pm0.47}$ | $77.36_{\pm0.66}$ | $81.92_{\pm0.97}$ | $54.24_{\pm0.73}$ |
| MemRL | API | $42.78_{\pm0.45}$ | $\mathbf{67.51}_{\pm0.60}$ | $9.27_{\pm0.38}$ | $52.67_{\pm0.72}$ | $44.53_{\pm0.54}$ | $58.71_{\pm0.86}$ | $45.91_{\pm0.59}$ |
| ReasoningBank | API | $46.11_{\pm0.91}$ | $67.01_{\pm0.55}$ | $7.49_{\pm0.91}$ | $52.33_{\pm1.09}$ | $70.15_{\pm0.70}$ | $79.48_{\pm0.76}$ | $53.76_{\pm0.82}$ |
| EvolveR | API | $44.44_{\pm0.45}$ | $64.99_{\pm0.73}$ | $6.42_{\pm0.25}$ | $53.00_{\pm1.00}$ | $79.85_{\pm0.93}$ | $83.33_{\pm0.10}$ | $55.34_{\pm0.58}$ |
| Llama-3.2-1B-Instruct | 1B | $43.33_{\pm0.78}$ | $61.62_{\pm0.24}$ | $7.85_{\pm0.29}$ | $53.00_{\pm0.94}$ | $62.19_{\pm0.20}$ | $73.76_{\pm0.06}$ | $50.29_{\pm0.33}$ |
| **UMEM-Llama-3.2-1B (Ours)** | 1B | $46.11_{\pm0.91}$ | $61.45_{\pm0.50}$ | $9.27_{\pm0.38}$ | $55.00_{\pm0.47}$ | $64.43_{\pm0.54}$ | $73.34_{\pm0.91}$ | $51.60_{\pm0.24}$ |
| Qwen3-4B-Instruct | 4B | $47.22_{\pm0.45}$ | $62.97_{\pm0.27}$ | $7.31_{\pm0.38}$ | $54.00_{\pm0.82}$ | $71.39_{\pm0.73}$ | $79.15_{\pm0.29}$ | $53.67_{\pm0.49}$ |
| **UMEM-Qwen3-4B (Ours)** | 4B | $52.22_{\pm0.45}$ | $65.66_{\pm0.63}$ | $8.38_{\pm0.15}$ | $54.67_{\pm0.27}$ | $\mathbf{83.09}_{\pm1.59}$ | $\mathbf{84.58}_{\pm1.15}$ | $58.10_{\pm0.71}$ |
| Qwen3-14B | 14B | $48.89_{\pm0.91}$ | $65.15_{\pm0.24}$ | $9.09_{\pm0.44}$ | $55.33_{\pm0.72}$ | $72.64_{\pm1.46}$ | $79.56_{\pm1.06}$ | $55.11_{\pm0.52}$ |
| **UMEM-Qwen3-14B (Ours)** | 14B | $\mathbf{55.56}_{\pm0.91}$ | $66.00_{\pm0.27}$ | $\mathbf{9.98}_{\pm0.52}$ | $\mathbf{55.67}_{\pm0.72}$ | $82.59_{\pm0.73}$ | $84.20_{\pm0.70}$ | $\mathbf{59.00}_{\pm0.31}$ |
| *Frozen Executor: Gemini-2.5-Flash* | | | | | | | | |
| No Memory | - | 53.33 | 73.23 | 10.16 | 30.00 | 55.22 | 72.89 | 49.14 |
| Self-RAG | API | $56.67_{\pm0.00}$ | $73.74_{\pm0.86}$ | $10.16_{\pm0.00}$ | $42.33_{\pm0.72}$ | $60.94_{\pm1.02}$ | $76.45_{\pm1.08}$ | $53.38_{\pm0.57}$ |
| ReMem | API | $57.78_{\pm0.91}$ | $69.87_{\pm0.50}$ | $10.70_{\pm0.50}$ | $45.67_{\pm0.27}$ | $55.72_{\pm0.41}$ | $73.05_{\pm1.06}$ | $52.13_{\pm0.16}$ |
| $Mem^p$ | API | $52.22_{\pm0.45}$ | $75.08_{\pm0.14}$ | $8.56_{\pm0.50}$ | $41.33_{\pm0.27}$ | $57.22_{\pm1.42}$ | $75.21_{\pm0.70}$ | $51.60_{\pm0.58}$ |
| MemRL | API | $49.45_{\pm1.20}$ | $70.37_{\pm0.36}$ | $10.87_{\pm1.16}$ | $42.67_{\pm0.88}$ | $57.96_{\pm0.89}$ | $78.69_{\pm0.71}$ | $51.67_{\pm0.87}$ |
| ReasoningBank | API | $56.11_{\pm1.20}$ | $75.09_{\pm0.27}$ | $11.23_{\pm0.50}$ | $46.33_{\pm0.72}$ | $59.95_{\pm0.20}$ | $77.65_{\pm0.65}$ | $54.39_{\pm0.59}$ |
| EvolveR | API | $57.78_{\pm0.55}$ | $75.25_{\pm0.63}$ | $11.05_{\pm0.14}$ | $39.33_{\pm0.54}$ | $55.22_{\pm0.35}$ | $75.87_{\pm0.62}$ | $52.42_{\pm0.47}$ |
| Llama-3.2-1B-Instruct | 1B | $52.78_{\pm0.91}$ | $73.06_{\pm0.14}$ | $9.45_{\pm0.29}$ | $42.00_{\pm0.00}$ | $52.99_{\pm0.35}$ | $72.76_{\pm1.59}$ | $50.51_{\pm0.33}$ |
| **UMEM-Llama-3.2-1B (Ours)** | 1B | $57.22_{\pm0.91}$ | $71.21_{\pm0.41}$ | $12.83_{\pm0.25}$ | $42.00_{\pm0.47}$ | $58.71_{\pm0.20}$ | $76.66_{\pm0.26}$ | $53.11_{\pm0.40}$ |
| Qwen3-4B-Instruct | 4B | $56.67_{\pm1.92}$ | $72.22_{\pm0.48}$ | $9.63_{\pm0.50}$ | $42.33_{\pm0.33}$ | $51.25_{\pm1.13}$ | $73.88_{\pm1.07}$ | $51.00_{\pm0.91}$ |
| **UMEM-Qwen3-4B (Ours)** | 4B | $58.33_{\pm0.78}$ | $\mathbf{75.93}_{\pm0.14}$ | $12.30_{\pm0.25}$ | $\mathbf{46.67}_{\pm1.20}$ | $61.69_{\pm1.77}$ | $77.86_{\pm0.77}$ | $55.46_{\pm0.82}$ |
| Qwen3-14B | 14B | $56.67_{\pm0.00}$ | $73.91_{\pm0.14}$ | $11.41_{\pm0.14}$ | $43.67_{\pm0.27}$ | $58.21_{\pm0.70}$ | $75.71_{\pm1.84}$ | $53.26_{\pm0.39}$ |
| **UMEM-Qwen3-14B (Ours)** | 14B | $\mathbf{58.89}_{\pm0.45}$ | $75.92_{\pm0.27}$ | $\mathbf{13.01}_{\pm0.38}$ | $45.67_{\pm0.54}$ | $\mathbf{63.19}_{\pm1.80}$ | $\mathbf{78.81}_{\pm2.12}$ | $\mathbf{55.91}_{\pm0.76}$ |

more powerful executors such as GPT-5.1 and Gemini-2.5-Flash tend to yield more pronounced gains compared to the Qwen3-8B-Thinking baseline. This phenomenon can be attributed to the higher-quality reasoning trajectories and interaction traces produced by stronger executors, which serve as high-fidelity source material for UMEM to distill more actionable and sophisticated insights.

Furthermore, UMEM exhibits excellent scalability regarding its policy model size. While even a compact 1B model (UMEM-Llama-3.2-1B) provides a substantial improvement over the base model and often surpasses larger models, further scaling the policy model to 4B and 14B generally yields additional performance dividends across nearly all tasks. This suggests that while UMEM is highly efficient at a small scale, increased model capacity allows it to capture more nuanced semantic relationships and implement more precise memory management strategies, thereby further pushing the performance upper bound of self-evolving agents.

*Table 2.* Ablation studies on joint optimization, reward design, neighborhood size, and embedding encoder on **GPT-5.1** and **Qwen3-8B-Thinking** under the default benchmark order. The full UMEM method is the baseline; blue ↓ and red ↑ mark relative drops and gains. The Δ Cost row reports token increases w/o $\mathcal{R}_{\text{eff}}$. Opt. denotes Optimization, and SNM denotes Semantic Neighborhood Modeling.

| Method | GPT-5.1 | | | | | | Qwen3-8B-Thinking | | | | | |
|---|---|---|---|---|---|---|---|---|---|---|---|---|
| | AIME | GPQA | HLE | HotpotQA | ALFWorld | | AIME | GPQA | HLE | HotpotQA | ALFWorld | |
| | (Acc.) | (Acc.) | (Acc.) | (Acc.) | SR | PR | (Acc.) | (Acc.) | (Acc.) | (Acc.) | SR | PR |
| *Joint Optimization Components* | | | | | | | | | | | | |
| **UMEM (Full Method)** | **51.67** | **65.15** | 8.56 | 54.00 | **82.84** | 84.20 | **58.33** | **53.54** | 8.02 | **63.00** | 50.75 | **73.13** |
| w/o Extraction Opt. | $45.00_{\downarrow 6.7}$ | $59.60_{\downarrow 5.6}$ | $5.88_{\downarrow 2.7}$ | $51.00_{\downarrow 3.0}$ | $76.12_{\downarrow 6.7}$ | $80.72_{\downarrow 3.5}$ | $55.00_{\downarrow 3.3}$ | $51.01_{\downarrow 2.5}$ | $8.02_{\uparrow 0.0}$ | $61.00_{\downarrow 2.0}$ | $45.53_{\downarrow 5.2}$ | $67.79_{\downarrow 5.3}$ |
| w/o Management Opt. | $48.33_{\downarrow 3.3}$ | $64.65_{\downarrow 0.5}$ | $9.09_{\uparrow 0.5}$ | $55.00_{\uparrow 1.0}$ | $80.60_{\downarrow 2.2}$ | $84.33_{\uparrow 0.1}$ | $56.67_{\downarrow 1.7}$ | $53.03_{\downarrow 0.5}$ | $6.95_{\downarrow 1.1}$ | $63.00_{\uparrow 0.0}$ | $44.70_{\downarrow 6.1}$ | $69.03_{\downarrow 4.1}$ |
| w/o SNM | $41.67_{\downarrow 10.0}$ | $64.14_{\downarrow 1.0}$ | $6.95_{\downarrow 1.6}$ | $52.00_{\downarrow 2.0}$ | $79.10_{\downarrow 3.7}$ | $81.09_{\downarrow 3.1}$ | $55.00_{\downarrow 3.3}$ | $50.00_{\downarrow 3.5}$ | $7.49_{\downarrow 0.5}$ | $60.00_{\downarrow 3.0}$ | $52.99_{\uparrow 2.2}$ | $72.30_{\downarrow 0.8}$ |
| *Effect of Efficiency Regularization* | | | | | | | | | | | | |
| w/o $\mathcal{R}_{\text{eff}}$ | $46.67_{\downarrow 5.0}$ | $61.11_{\downarrow 4.0}$ | $9.63_{\uparrow 1.1}$ | $58.00_{\uparrow 4.0}$ | $78.36_{\downarrow 4.5}$ | $83.08_{\downarrow 1.1}$ | $56.67_{\downarrow 1.7}$ | $50.00_{\downarrow 3.5}$ | $7.49_{\downarrow 0.5}$ | $60.00_{\downarrow 3.0}$ | $50.00_{\downarrow 0.8}$ | $72.39_{\downarrow 0.7}$ |
| Δ Cost (Token) | +23.57 | +5.49 | +0.98 | +4.36 | +112.00 | | +486.00 | +148.99 | +337.00 | +603.00 | +973.00 | |
| *Sensitivity to Neighborhood Size* | | | | | | | | | | | | |
| **UMEM (N = 3, Ours)** | **51.67** | **65.15** | 8.56 | **54.00** | **82.84** | **84.20** | **58.33** | **53.54** | 8.02 | **63.00** | 50.75 | 73.13 |
| N = 1 (Too Narrow) | $48.33_{\downarrow 3.3}$ | $63.13_{\downarrow 2.0}$ | $7.49_{\downarrow 1.1}$ | $50.00_{\downarrow 4.0}$ | $78.36_{\downarrow 4.5}$ | $81.34_{\downarrow 2.9}$ | $51.67_{\downarrow 6.7}$ | $51.52_{\downarrow 2.0}$ | $8.02_{\uparrow 0.0}$ | $59.00_{\downarrow 4.0}$ | $41.79_{\downarrow 9.0}$ | $67.66_{\downarrow 5.5}$ |
| N = 5 (Too Broad) | $46.67_{\downarrow 5.0}$ | $64.65_{\downarrow 0.5}$ | $7.49_{\downarrow 1.1}$ | $52.00_{\downarrow 2.0}$ | $81.34_{\downarrow 1.5}$ | $84.08_{\uparrow 0.1}$ | $58.33_{\uparrow 0.0}$ | $52.53_{\downarrow 1.0}$ | $9.63_{\uparrow 1.6}$ | $62.00_{\downarrow 1.0}$ | $42.54_{\downarrow 8.2}$ | $73.26_{\uparrow 0.1}$ |
| *Sensitivity to Embedding Model* | | | | | | | | | | | | |
| **UMEM (BGE-M3, Ours)** | **51.67** | 65.15 | 8.56 | 54.00 | 82.84 | 84.20 | 58.33 | 53.54 | 8.02 | 63.00 | 50.75 | 73.13 |
| Qwen3-Embedding-0.6B | $50.00_{\downarrow 1.7}$ | $65.66_{\uparrow 0.5}$ | $9.09_{\uparrow 0.5}$ | $56.00_{\uparrow 2.0}$ | $83.58_{\uparrow 0.7}$ | $85.82_{\uparrow 1.6}$ | $58.33_{\uparrow 0.0}$ | $54.04_{\uparrow 0.5}$ | $7.49_{\downarrow 0.5}$ | $63.00_{\uparrow 0.0}$ | $50.75_{\uparrow 0.0}$ | $73.01_{\downarrow 0.1}$ |

## 5.3. Ablation Studies

This section validates the effectiveness of our designs in UMEM by ablation studies: (1) the necessity and sensitivity of Semantic Neighborhood Modeling, including neighborhood size and embedding encoder choice; (2) joint optimization of memory extraction and management; and (3) the contribution of efficiency regularization to both performance and inference cost.

**Semantic Neighborhood Modeling.** We first examine the necessity of Semantic Neighborhood Modeling. The w/o SNM row in Table 2 reveals that removing it during training results in significant performance collapse, particularly on the reasoning-heavy AIME benchmark (GPT-5.1: dropping from 51.67 to 41.67; Qwen3-8B: dropping from 58.33 to 55.00). Furthermore, we investigate the impact of the semantic neighborhood size $N \in \{1, 3, 5\}$. As reported in the Sensitivity to Neighborhood Size block of Table 2, $N = 3$ yields the optimal balance between task-specific optimization and cross-task transfer. Performance degrades at both extremes: an overly narrow neighborhood ($N = 1$) fails to capture task shifts (GPT-5.1: AIME drops to 48.33; Qwen3-8B: dropping from 58.33 to 51.67), while an overly broad one ($N = 5$) introduces noise that dilutes the reward signal during optimization. Finally, we evaluate the sensitivity to the embedding encoder used for neighborhood construction. Replacing BGE-M3 with Qwen3-Embedding-0.6B yields comparable average performance on both GPT-5.1 (57.74 vs. 58.36) and Qwen3-8B-Thinking (51.13 vs. 51.10), suggesting that UMEM is robust to encoder choice when the neighborhood remains a reliable proxy for future reuse.

**Joint Optimization.** We evaluate the contribution of memory extraction and management by masking the "gradient" of their respective tokens. As shown in the first two rows of Table 2, breaking the joint optimization leads to severe performance degradation across the majority of benchmarks. Specifically, disabling memory extraction optimization results in an average performance decline of 3.88 points across all metrics, which is significantly higher than that observed when removing management optimization (1.48 points). These results reveal that optimizing the quality of extracted memory is the more important for effective self-evolution.

**Efficiency Regularization.** We further investigate the role of the efficiency regularization term $\mathcal{R}_{\text{eff}}$. As reported in Table 2, removing $\mathcal{R}_{\text{eff}}$ decreases the average performance from 57.74 to 56.14 on GPT-5.1 and from 51.13 to 49.43 on Qwen3-8B-Thinking. This trend holds across both non-thinking execution (GPT-5.1 with reasoning set to `none`) and thinking-mode execution (Qwen3-8B-Thinking). Meanwhile, the Δ Cost row shows that token consumption consistently increases after removing this term, with particularly large increases on Qwen3-8B-Thinking (e.g., +603 tokens on HotpotQA and +973 tokens on ALFWorld). These results suggest that $\mathcal{R}_{\text{eff}}$ guides the Mem-Optimizer toward memory update actions that improve downstream performance while reducing redundant executor reasoning.

## 5.4. Cross-Task Generalization

We evaluate cross-task memory transfer to test whether evolved memories remain useful beyond the benchmark stream from which they are extracted. Specifically, we first build the memory bank on source benchmarks and then evaluate the executor on out-of-distribution (OOD) target tasks using the transferred memories. All results are averaged over three randomized task sequences to reduce ordering effects. As shown in Table 3, UMEM-Qwen3-4B

*Table 3.* Cross-task generalization averaged over three randomized task sequences. **CTU** is OOD accuracy using memories from different prior benchmarks; **ERR** is the percentage of no-memory failures resolved by transferred memory.

| Method | CTU (%) | ERR (%) |
|---|---|---|
| ReasoningBank | 18.20 | 11.60 |
| EvolveR | 30.30 | 15.10 |
| $Mem^p$ | 37.20 | 16.00 |
| No-Train-4B | 39.29 | 17.80 |
| MemRL | 40.40 | 17.30 |
| **UMEM-4B (Ours)** | **41.47** | **23.17** |

achieves the best performance on both CTU (41.47%) and ERR (23.17%). The gains over No-Train-4B (2.18 points on CTU and 5.37 points on ERR) indicate that the improvement comes from optimized memory evolution rather than the extraction prompt alone. Among strong memory baselines, MemRL obtains the closest CTU (40.40%), yet its ERR remains much lower than UMEM (17.30% vs. 23.17%). This indicates that MemRL's reranking-based memory selection can reduce harmful transfer, but its transferred memories are less effective at recovering failures on OOD tasks. In contrast, UMEM improves both aggregate OOD accuracy and error recovery, showing that neighborhood-level optimization yields more transferable evolved memories.

### 5.5. Stability of Self-Evolution

We evaluate UMEM under a continual learning setting across both single-turn reasoning benchmarks and the multi-turn ALFWorld environment, reporting the cumulative accuracy in Figure 3. In this streaming protocol, the agent must continuously evolve its memory bank without resetting. This poses a severe challenge: error accumulation. As interaction proceeds, flawed memory extraction policies tend to pollute the memory bank with noise or instance-specific shortcuts, degrading performance on subsequent tasks. As shown in Figure 3, under this challenging setting, UMEM consistently maintains a superior performance curve compared to baselines, particularly in the later stages. It exhibits significantly slower and more controlled degradation than ReMem and MemP across all evaluations, with the performance gap widening as interaction proceeds. Crucially, ReMem (green curve), which optimizes memory management in isolation, suffers the most rapid degradation and results in the lowest final performance, proving the necessity of joint optimization. This behavior indicates that UMEM accumulates fewer harmful memories over long horizons, and that its advantage stems not from whether memory is learned, but from how memory extraction and management are coordinated during continual evolution.

The extracted memories of the baselines like ReMem and $Mem^p$ may appear locally better, yet their long-horizon utility remains opaque to the memory manager. Consequently,

such memories are often retained and repeatedly reused even when they introduce subtle reasoning errors, leading to progressive error amplification in cumulative evaluation. In contrast, the substantially reduced degradation observed for UMEM suggests that newly updated memories are more consistently aligned with future reuse.

Taken together, these results support the conclusion that stable self-evolution requires memory updates to be tightly coupled with the context in which errors arise. By evolving memory primarily around experiences most relevant to the current trajectory, UMEM promotes structured knowledge consolidation rather than unconstrained accumulation. From an optimization perspective, this behavior corresponds to sparse, localized updates over external memory parameters, which naturally limit interference and mitigate long-horizon error accumulation.

### 5.6. Test-Time Self-Evolution

To further validate the sustainability of self-evolution beyond the single-epoch setting in Section 5.5, we extend the experimental scope from 1 epoch to a rigorous 10-epoch long-horizon continual interaction on the ALFWorld benchmark with GPT-5.1 as the executor. Figure 4 reports both epoch-wise and cumulative Success Rate and Progress Rate. As shown in the per-epoch Success Rate, UMEM consistently achieves the highest performance across all epochs. Although online retrieval and memory updates inevitably introduce performance fluctuations, UMEM recovers quickly after temporary drops, indicating a well-balanced memory strategy between exploration and stability during continual evolution. The cumulative Success Rate further highlights UMEM's advantage. UMEM shows a steady and sustained improvement trend, converging to a substantially higher performance level than all baselines. Beyond final task success, UMEM also consistently outperforms baselines on Progress Rate, with a particularly pronounced margin in cumulative metrics. This trend suggests that, even in partially unsuccessful episodes, UMEM tends to execute more correct intermediate steps, reflecting more stable multi-step decision-making. Overall, these results indicate that UMEM supports a more stable and sustainable form of agent self-evolution under continual interaction.

### 5.7. Cross-Model Effectiveness and Efficiency

Figure 5 reports Success Rate and Average Steps on ALFWorld across different executor LLMs. UMEM consistently achieves the highest Success Rate for all executors, indicating that the evolved experiences provide robust, executor-agnostic performance gains. Notably, this improvement is accompanied by a clear reduction in Average Steps, showing that higher success is not obtained through longer or more exploratory interaction trajectories, but through more

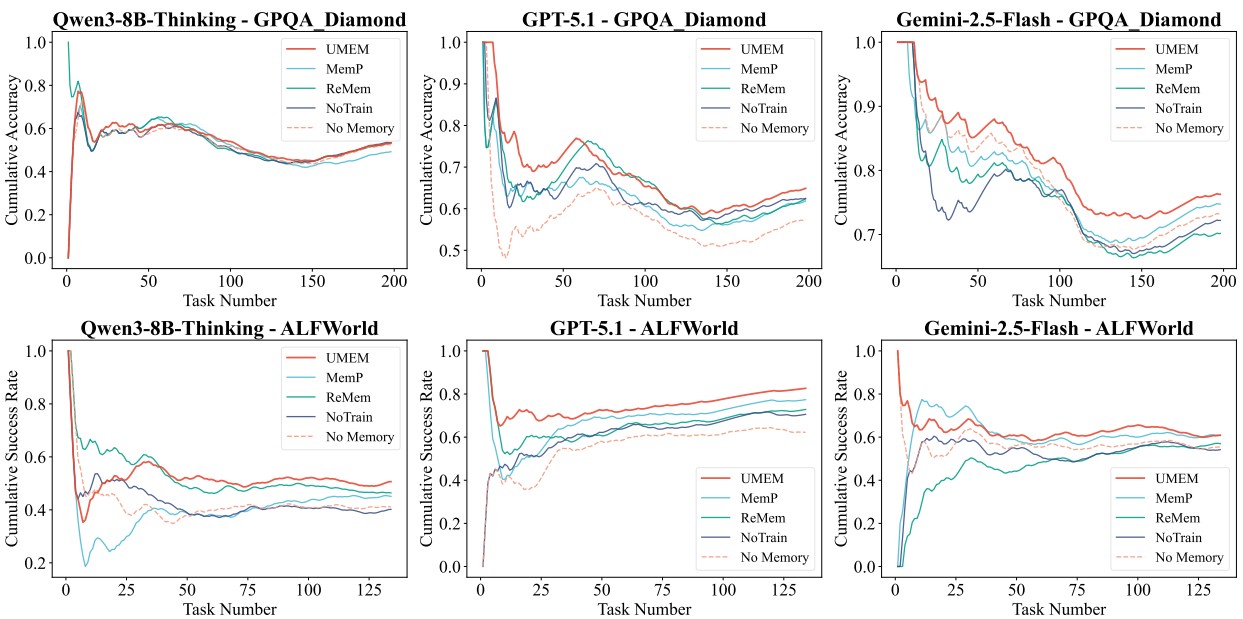

*Figure 3.* Cumulative performance over sequential tasks on GPQA-Diamond and ALFWorld Benchmarks.

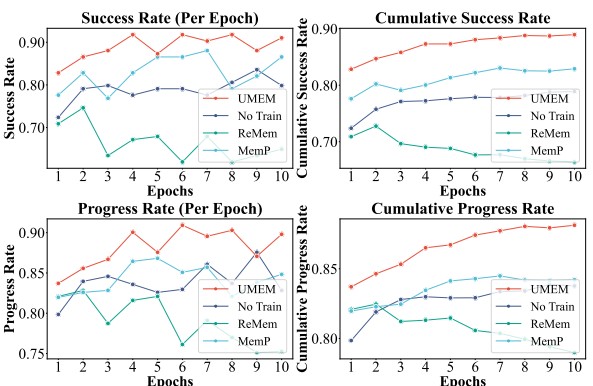

*Figure 4.* Test-Time Self-Evolution on ALFWorld.

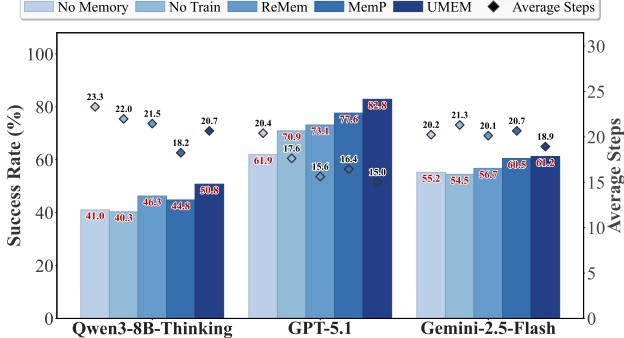

*Figure 5.* Success Rate and Average Steps on ALFWorld benchmark across different executor models.

efficient decision making during interaction. This efficiency gain is evident in case study 8.

The joint improvement in success and efficiency provides insight into the nature of the experiences evolved by UMEM. In long-horizon interactive tasks, overly specific experiences often lead to shortcut behaviors that fail to generalize to similar tasks, ultimately causing execution failures; in contrast, overly coarse heuristics fail to sufficiently constrain execution and result in longer trajectories. Across all executor models, UMEM consistently avoids these failure modes, achieving higher success with fewer execution steps. This pattern indicates that the observed gains reflect a genuine improvement in execution efficiency that generalizes across executors, rather than an artifact of increased interaction length or model-specific behavior.

## 6. Conclusion

In this paper, we introduced UMEM for self-evolving agents. Unlike prior approaches that treat memory extraction and management as static or decoupled processes, UMEM achieves joint optimization of extraction and management through Semantic Neighborhood Modeling and GRPO augmented with a Marginal Utility Reward. This design effectively mitigates the accumulation of instance-specific noise and ensures that extracted memories are intrinsically aligned with the agent's management policy. Empirical results demonstrate that UMEM significantly outperforms highly competitive baselines in both cross-task generalization and execution efficiency. By enabling agents to continuously refine the memory bank during continuous interaction, UMEM offers a robust paradigm for realizing lifelong learning in open-ended environments.

## Impact Statement

This work aims to advance self-evolving agents by improving how frozen executors reuse external memories without parameter updates. It may reduce the cost of continual adaptation, but persistent memory also introduces potential misuse risks. For example, an attacker may craft query sequences that inject incorrect memories into the memory bank and influence the agent's subsequent decisions. Practical applications should therefore incorporate memory verification and adversarial robustness training before using self-evolving memory systems in high-stakes settings.

## Acknowledgment

This work is supported by the National Science and Technology Major Project (Grant No. 2022ZD0116101), the National Natural Science Foundation of China (NSFC) under Grant No. 62206295, the Major Scientific Research Project of the State Language Commission in the 13th Five-Year Plan (Grant No. WT135-38), the public technology service platform project of Xiamen City (No. 3502Z20231043).

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

## A. Limitations

UMEM improves self-evolution by extracting memories that can be reused by future queries, but such reuse depends on whether relevant memories are retrieved. Therefore, the utility of generalized memories is constrained by semantic similarity between source and target tasks and by the quality of the embedding encoder. Although our ablations show robustness to the tested encoders, biased or poorly calibrated encoders may still introduce retrieval bias into the memory bank. Moreover, transfer utility is not strictly monotonic under distribution shift: when source and target tasks are weakly related, retrieved memories may provide limited benefit or even interfere with execution. Memory should also be a selective aid rather than a default mechanism; future work may improve retrieval and invocation policies so that the executor can better decide when to use external memory.

## B. Implementation Details

**UMEM Implementation.** We optimize the Mem-Optimizer using GRPO (Shao et al., 2024). For each update, we sample a batch of 128 training queries and generate $G=8$ candidate memory actions per query. Training is conducted for 3 epochs. Semantic neighborhoods are constructed using BGE-M3 (Chen et al., 2024) as the embedding encoder and cosine similarity as the retrieval metric, with Top-$N=3$ neighbors for neighborhood-level reward evaluation. During memory-augmented execution and online memory evolution, the memory bank is queried with the same encoder and retrieval metric, and the Top-$K=3$ memories are provided to the frozen executor as context. We apply KL regularization with coefficient $\beta=0.001$ and use a clipping ratio of $\epsilon=0.2$. The learning rate is set to $1 \times 10^{-6}$. During Mem-Optimizer training, generation is performed with temperature 1.0 to encourage exploration, while executor inference and all evaluation runs use greedy decoding with temperature 0.0. Our method is implemented using the `verl` framework (Sheng et al., 2025) and trained on 16 NVIDIA A100 (80GB) GPUs for approximately 14 hours, where 8 GPUs are used to train the Mem-Optimizer and the remaining 8 GPUs are dedicated to serving the executor model (Qwen3-8B-Thinking) for trajectory generation during training.

**Baseline Implementation.** To ensure a fair comparison, all baselines are evaluated under the same streaming protocol as UMEM, where benchmark instances are processed sequentially and the memory bank is updated online without resetting within the same stream. All methods use the same frozen executor LLMs, executor prompt templates, benchmark order, output parsing rules, and decoding temperature of 0.0. For memory-based baselines, we standardize the retrieval backend by using BGE-M3 embeddings, cosine-similarity retrieval, and Top-$K=3$ retrieved memories as execution context. We impose no explicit memory-size constraint on any method, resulting in an unlimited memory budget for all baselines. For all baselines, including Self-RAG (Asai et al., 2024), ReMem (Wei et al., 2025), $Mem^p$ (Fang et al., 2025), ReasoningBank (Ouyang et al., 2025), MemRL (Zhang et al., 2026), and EvolveR (Wu et al., 2025), we follow the hyperparameter settings, extraction prompts, memory management rules, and update procedures described in their original papers whenever applicable. The No Memory baseline disables retrieval and memory updates entirely, while the No Train baseline uses the same Mem-Optimizer prompt template and memory update interface as UMEM but does not apply GRPO optimization.

## C. Mem-Optimizer Action Template

Each Mem-Optimizer action is represented as a structured output following the template:

```
<experience><value>...</value><operation>...</operation></experience>
```

where `<value>` encodes the extracted memory content derived from the interaction trace, and `<operation>` specifies the corresponding memory evolution decision (e.g., addition, replacement).

## D. Theoretical Analysis

### D.1. Cosine Neighborhood as a Proxy for Reuse-Semantic Proximity

**Lemma D.1** (Retrieval-score stability under cosine proximity). *Let $e(\cdot)$ be $\ell_2$-normalized embeddings, i.e., $\|e(x)\|_2 = 1$. For any two queries $q_1, q_2$ and any candidate key $k$,*

$$\left| e(q_1)^\top e(k) - e(q_2)^\top e(k) \right| \leq \|e(q_1) - e(q_2)\|_2 = \sqrt{2 - 2\, e(q_1)^\top e(q_2)}.$$

---

**Algorithm 1:** UMEM Training: Semantic Neighborhood Modeling and GRPO

---

**Input:** Query corpus $\mathcal{D}$, frozen executor $\mathcal{E}$, Mem-Optimizer $\pi_\phi$, Neighborhood size $N$, Group size $G$
**Output:** Trained parameters $\phi$ and evolved memory bank $\mathcal{B}$

1   **Phase 1: Offline Semantic Neighborhood Modeling**; **foreach** $q \in \mathcal{D}$ **do**
2       $\mathcal{N}_N(q) \leftarrow$ Retrieve $N$ nearest neighbors for $q$ from $\mathcal{D} \setminus \{q\}$;

3   **Phase 2: GRPO-based Online Memory Evolution**; **for** *each training step* **do**
4       Sample a mini-batch $\mathbf{Q} \subset \mathcal{D}$; **foreach** $q \in \mathbf{Q}$ **do**
5           $\mathcal{B}^{topk} \leftarrow$ Retrieve Top-$K$ memories from $\mathcal{B}$ by cosine similarity with $e(q)$; $\tau_q \leftarrow \mathcal{E}(q, \mathcal{B}^{topk})$; **for** $g \leftarrow 1$ **to** $G$ **do**
6               $o^{(g)} \sim \pi_\phi(\cdot \mid q, \tau_q, \mathcal{B}^{topk})$; $r_f^{(g)} \leftarrow \mathbb{I}[\text{FormatOK}(o^{(g)})]$; $\tilde{\mathcal{B}}^{(g)} \leftarrow$ Apply $o^{(g)}$ to $\mathcal{B}$;
                $r_g^{(g)} \leftarrow \frac{1}{|\mathcal{N}_N(q)|} \sum_{q' \in \mathcal{N}_N(q)} \text{UtilityGain}(q', \tilde{\mathcal{B}}^{(g)}, \mathcal{B})$; $r^{(g)} \leftarrow r_f^{(g)} + r_g^{(g)}$;
7           Update $\phi$ via GRPO using group advantages $\{r^{(g)} - \text{mean}(r)\}_{g=1}^{G}$; $\mathcal{B} \leftarrow \tilde{\mathcal{B}}^{(g^\star)}$ where $g^\star = \arg\max_g r^{(g)}$;

8   **return** $\phi, \mathcal{B}$;

---

*Proof.* Since $\|e(k)\|_2 = 1$, by Cauchy–Schwarz,

$$\left| e(q_1)^\top e(k) - e(q_2)^\top e(k) \right| = \left| (e(q_1) - e(q_2))^\top e(k) \right| \leq \|e(q_1) - e(q_2)\|_2.$$

For unit vectors, $\|u - v\|_2^2 = 2 - 2u^\top v$, hence the equality. $\qquad\square$

**Interpretation.** High cosine similarity guarantees that $q_1$ and $q_2$ assign nearly identical relevance scores to any memory key. This score stability ensures highly overlapping retrieval rankings (and thus similar Top-$K$ sets). Consequently, the cosine neighborhood of a source query effectively captures the cluster of future queries that will likely retrieve (and reuse) the same memory.

## E. Procedure for Evolutionary Memory Management

Algorithm 1 details the training process of UMEM, characterized by the co-evolution of the Mem-Optimizer $\pi_\phi$ and the memory bank $\mathcal{B}$. Prior to training, we perform **Semantic Neighborhood Modeling** to identify $\mathcal{N}_N(q)$ for each query $q$ based on embedding similarity, preventing shortcut learning. The Mem-Optimizer is then optimized through the following iterative stages:

- **(1) Memory-Augmented Execution**: The frozen executor $\mathcal{E}$ performs task $q$ using retrieved context from the current memory $\mathcal{B}$ to generate an initial trajectory $\tau_q$.

- **(2) Policy Rollout**: The Mem-Optimizer $\pi_\phi$ samples a group of $G$ candidate operations $\{o^{(g)}\}_{g=1}^{G}$ (e.g., ADD or UPDATE) based on $q$, $\tau_q$, and the retrieved memory.

- **(3) Marginal Utility Reward**: For each rollout, we compute a format reward $r_f$ for structural correctness and a marginal utility reward $r_g$, defined as the average performance gain (success rate and efficiency) across the semantic neighborhood $\mathcal{N}_N(q)$.

- **(4) Optimization via GRPO**: The policy $\pi_\phi$ is updated using group-relative advantages derived from the combined rewards $r_f + r_g$, facilitating stable policy refinement without a critic network.

- **(5) Online Memory Evolution**: The memory bank $\mathcal{B}$ is updated by committing the best-performing operation $o^{(g^\star)}$ from the group, ensuring the knowledge base evolves alongside the policy.

## F. Training Diagnostics and Forgetting Analysis

We analyze whether UMEM training introduces two potential risks: low-quality memory generation by the Mem-Optimizer and catastrophic forgetting caused by continual GRPO optimization.

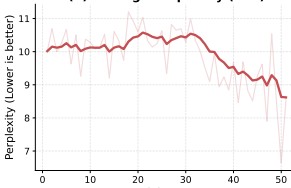 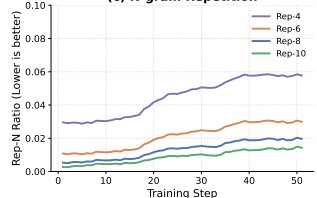 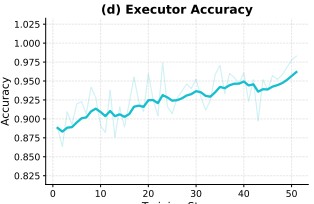

*Figure 6.* Training diagnostics during UMEM optimization.

*Table 4.* Forgetting analysis on instruction following and code generation.

| Model | IFEval (Strict Avg.) | HumanEval (Pass@1) |
|---|---|---|
| Qwen3-4B | 85.53% | 63.53% |
| **UMEM-4B** | **87.33%** | **65.85%** |

**Training mechanism.** Small Mem-Optimizers can produce lower-quality memories, but in UMEM these errors are actively corrected during training rather than simply accumulated: First, under GRPO, low-quality memories receive low marginal utility reward because they fail to provide valuable feedback to the semantic neighborhood, so the policy is directly pushed away from such generations. Second, UMEM's online UPDATE mechanism can revise previously stored low-quality memories within the current Top-$K$ retrieved set, instead of leaving them in the memory bank unchanged. This is exactly the motivation of neighborhood-level reward and joint extraction-management optimization in UMEM.

**Empirical diagnostics.** As shown in Figure 6, we track several training diagnostics, which collectively show that the Mem-Optimizer is optimized to generate more fluent and reusable memory content for future tasks: PPL decreases from 10.02 to 8.56, format reward rises from 0.81 to 1.00 (reaching 0.95 by step 2), 10-gram repetition stays low (average 0.0097), and execution accuracy over semantic neighborhoods improves by about 9%. These results suggest that low-quality memories are effectively corrected during UMEM training, even with a small model.

We do not observe evidence that UMEM induces catastrophic forgetting. As shown in Figure 6, execution accuracy over semantic neighborhoods increases by about 9% during training, indicating that memory quality improves rather than degrades. Moreover, our main evaluations are on out-of-distribution tasks; if the learned memories had deteriorated, such gains would be unlikely to persist. In addition, UMEM uses RL-based optimization, which updates only a small fraction of parameters (5.89% above the $1 \times 10^{-5}$ threshold). Finally, Table 4 evaluates potential catastrophic forgetting on IFEval and HumanEval, where UMEM-4B improves over Qwen3-4B on both benchmarks.

## G. Additional Training Cost Analysis

We provide an itemized wall-clock breakdown of UMEM training cost and a scaling analysis with respect to the semantic neighborhood size $N$ and the number of GRPO rollouts $G$ in Table 5. All results are profiled on $8\times$ NVIDIA A100 (80GB) GPUs under the anchor setting used in Appendix B, namely $N=3$, $K=3$, and $G=8$. Under this setting, UMEM takes 949.60 seconds (15.83 minutes) per optimization step on average.

**Per-step executor complexity.** The total number of executor calls per training step is

$$E + 2GN,$$

where $E=1$ denotes the initial executor rollout used for memory extraction, and $2GN$ corresponds to reward evaluation for $G$ candidate memory updates over $N$ semantic neighbors under both *with-memory* and *without-memory* conditions. This makes reward evaluation the dominant training bottleneck. We view this overhead as a necessary trade-off for optimizing memory updates based on neighborhood-level utility rather than single-instance feedback.

*Table 5.* Training cost analysis. The default setting uses $N=3$, $K=3$, and $G=8$ and is profiled on $8\times$ NVIDIA A100 (80GB) GPUs. The left block reports the per-step wall-clock breakdown and share of total runtime; the right blocks report average seconds per optimization step when scaling the semantic neighborhood size $N$ or the number of GRPO rollouts $G$.

| Subtask | Default | | Scale with $N$ ($G=8$) | | | Scale with $G$ ($N=3$) | | |
|---|---|---|---|---|---|---|---|---|
| | Time | Share | $N=1$ | $N=3$ | $N=5$ | $G=4$ | $G=8$ | $G=16$ |
| Executor rollout | 175.90 | 18.5% | 182.70 | 175.90 | 171.00 | 199.30 | 175.90 | 183.00 |
| Mem-Optimizer rollout | 27.50 | 2.9% | 26.20 | 27.50 | 29.80 | 22.10 | 27.50 | 40.70 |
| Reward evaluation | 623.90 | 65.7% | 316.00 | 623.90 | 970.80 | 364.10 | 623.90 | 1434.10 |
| GRPO optimization | 108.00 | 11.4% | 101.80 | 108.00 | 116.20 | 52.60 | 108.00 | 227.20 |
| Memory update | 0.80 | 0.1% | 0.50 | 0.80 | 0.70 | 0.60 | 0.80 | 0.70 |
| Other overhead | 13.50 | 1.4% | 0.00 | 13.50 | 0.00 | 0.00 | 13.50 | 0.00 |
| Total | 949.60 | 100.0% | 627.30 | 949.60 (*ours*) | 1288.40 | 638.70 | 949.60 (*ours*) | 1885.60 |

### G.1. Itemized Breakdown of Training Cost

The left block of Table 5 presents the average runtime of each subtask under the default setting. The primary bottleneck is reward evaluation, which accounts for 65.7% of the total training time. This is mainly caused by the $G$ GRPO rollouts and the associated executor executions required to compute neighborhood-level rewards. In contrast, memory update itself is negligible, accounting for only 0.1% of the total runtime.

### G.2. Scaling Analysis

We next analyze how the main hyperparameters affect training cost. The right blocks of Table 5 show scaling with the semantic neighborhood size $N$ while fixing $E=1$ and $G=8$, and scaling with the number of GRPO rollouts $G$ while fixing $E=1$ and $N=3$. As expected, increasing either $N$ or $G$ leads to an approximately linear increase in reward-evaluation cost, and also increases GRPO optimization time. By contrast, increasing $K$ only slightly extends the prompt context and has minimal runtime impact, since it does not introduce additional executor calls.

These results show that the dominant cost of UMEM comes from executor-based reward evaluation rather than memory update itself. In practice, $N=3$ and $G=8$ provide a reasonable trade-off between training cost and performance, which is why we adopt them in our main experiments.

## H. Prompt Templates

We present the detailed instruction templates used in our framework, encompassing both the Memory Optimizer and the Executor LLM. First, the system prompt for the Memory Optimizer, which is responsible for refining and organizing retrieved past experiences, is shown in Prompt 7. For the Executor LLM, we designed distinct system prompts to adhere to specific output formats across different domains during training and evaluation. Specifically, mathematical reasoning tasks follow the instructions in Prompt J.1. The unified template for multiple-choice questions (handling both index-based and letter-based outputs) is presented in Prompt J.2, while general question-answering tasks are guided by Prompt I.1.

## I. Case Study

Figure 8 shows how retrieved experiences enable effective knowledge transfer and task completion. The task "put a clean cloth in countertop" contains an implicit requirement: the cloth must be *cleaned* before placement, not merely moved.

**UMEM Enhanced Agent.** By retrieving experiences from analogous tasks (cleaning plates, knives, and pans), the agent recognizes a generalizable pattern: *locate object $\rightarrow$ pick up $\rightarrow$ go to sinkbasin $\rightarrow$ clean with sinkbasin $\rightarrow$ place on target*. Although the agent initially explores incorrect locations (handtowelholder) and picks up the wrong object (handtowel), it self-corrects upon discovering the cloth and successfully applies the cleaning procedure learned from memory. This demonstrates the agent's ability to **transfer procedural knowledge** across different object types (plate/knife/pan $\rightarrow$ cloth) and **recover from exploration errors** through experience-guided reasoning.

**Baseline.** Lacking prior experiences, this agent interprets the task literally as a simple pick-and-place operation. Despite

locating the cloth quickly, it repeatedly executes `take → move` actions without ever invoking the `clean` command. Notably, even after querying the `help` command and seeing "clean (object) with (receptacle)" in the available actions, the agent fails to connect this capability to the task requirement. This reveals a critical limitation: **without experiential knowledge linking the task semantics to the required action sequence, the agent cannot infer the missing step**, resulting in an ineffective loop of 30 repeated attempts.

**Key Insights.** (1) *Semantic understanding*: Experiences provide crucial context for interpreting implicit task requirements ("clean" as a prerequisite, not just a descriptor). (2) *Efficiency*: With memories extracted by UMEM, the executor completes the task in 13 steps through meaningful exploration, whereas the baseline agent falls into a futile loop of repetitive actions and exhausts 30 steps without solving the task. (3) *Generalization*: Experiences about cleaning plates/knives/pans successfully transfer to cleaning cloth, demonstrating cross-object procedural generalization.

---

> **Prompt I.1: Executor Prompt Template: Question Answering (QA)**
>
> **[System]:** `# Role`
> You are an expert Question Answering Agent. Your goal is to answer questions based on the provided **Context** and applying **Past Effective Experiences**.
> `# Input Data`
> 1. **Past Experiences**: Historical context, strategies, or rules to guide your reasoning.
> 2. **Context**: Background information, documents, or text passages relevant to the question.
> 3. **Question**: The specific inquiry you need to answer.
> `# Instructions`
> 1. Read the **Context** carefully to extract relevant facts.
> 2. Refer to **Past Experiences** to find successful reasoning patterns or specific knowledge that supplements the context.
> 3. Synthesize the information to answer the **Question** accurately and concisely.
>
> - - - - - - - - - - - - - - - - - - - - - - - - - - - - - - - - - - - - - - - - - - - - - - - - - - - - - - - - - - - - - - -
>
> **[User]:** `# Current Task`
> **Question**:
> `{question}`
> `{context_block}`
> **Past Experiences**:
> `{memory_section}`
> `# Output Format`
> You must strictly follow this format:
> First, provide your reasoning process, citing the context or experiences where applicable.
> Then, output the final answer wrapped in `\boxed{}`.

---

## J. Failure Analysis

While scaling models isstability, failure analysis clarifies understanding UMEM's limitations. We examine examine two failure modes: cases where UMEM fails but No Memory or No Train succeeds. Though infrequent (accounting for ∼7% of cases in UMEM-1B and ∼5% in UMEM-4B), these errors reveal two key issues:

**Memory Interference (No Memory correct, UMEM fails).** When the prompt already contains all necessary information (e.g., extractive QA or self-contained reasoning), injecting external memory can distract the executor. This suggests memory usage should not be forced by default. One possible direction is to encapsulate memory retrieval as a callable tool, allowing the executor to decide whether to invoke it based on the task.

**Abstraction vs. Precision Trade-off (No Train correct, UMEM fails).** By filtering surface details, UMEM prioritizes broad generalization over rote memorization. As a result, No Train occasionally outperforms it on specific queries requiring exact instance-specific constraints (e.g., isolated ALFWorld instances) because it preserves unabstracted raw data. Furthermore, extracting raw experiences into generalizable rules is inherently challenging; consequently, a smaller Mem-Optimizer such as UMEM-1B struggles more with this delicate process, introducing slightly more extraction imperfections.

Overall, these failures show that memory should be a selective aid rather than a default mechanism. The core challenge lies

in dynamically determining not only whether to invoke memory, but also the appropriate level of abstraction for a given task.

---

**Prompt J.1: Executor Prompt Template: Mathematical Reasoning**

**[System]:** `# Role`
You are an expert Math Task Execution Agent. Your goal is to solve mathematical problems by applying logic and methods from **Past Effective Experiences**.
`# Input Data`
1. **Past Experiences**: Relevant formulas, theorems, or similar solved examples.
2. **Question**: The specific math problem you need to solve.
`# Instructions`
1. Analyze the **Question** to identify the mathematical concepts involved.
2. Refer to the **Past Experiences** to find the correct formula, method, or logic pattern.
3. Perform the **Problem Solving Process** step-by-step. Show your work, calculations, and derivations clearly.

- - - - - - - - - - - - - - - - - - - - - - - - - - - - - - - - - - - - - - - - - - - -

**[User]:** `# Current Task`
Solve the following problem.
**Question**:
{question}
**Past Experiences**:
{memory_section}
Please reason step by step, and put your final answer within \boxed{}.

---

**Prompt J.2: Executor Unified Prompt Template: Multiple Choice Tasks**

**[System]:** `# Role`
You are an expert Task Execution Agent. Your goal is to solve multiple-choice questions by applying **Past Effective Experiences**.
`# Input Data`
1. **Past Experiences**: Historical context or rules to guide your decision.
2. **Question**: The specific problem you need to solve.
3. **Options**: A list of candidate answers.
`# Instructions`
1. Analyze the **Question** carefully.
2. Refer to the **Past Experiences** to find the logic or evidence required to solve the problem.
3. Evaluate the **Options** and select the best one.
4. **CRITICAL**:

  - *[For Index Tasks]:* Identify the **Index** of the selected option based on a **0-based system** (i.e., 0, 1, ...).

  - *[For Letter Tasks]:* Identify the **Letter** of the selected option (i.e., A, B, C, or D).

- - - - - - - - - - - - - - - - - - - - - - - - - - - - - - - - - - - - - - - - - - - -

**[User]:** `# Current Task`
**Question**:
{question}
**Options**:
{choice_block}
**Past Experiences**:
{memory_section}
`# Output Format`
Analyze the options and the question step-by-step.
Output the final answer

  - *[For Index Tasks]:* index wrapped in \boxed{index}, e.g., \boxed{0}.

  - *[For Letter Tasks]:* single letter wrapped in \boxed{Letter}, e.g., \boxed{A}.

---

**Mem-Optimizer Prompt: Success Case (Training & Evaluation)**

# System Prompt
# Role
You are an expert **Experience Summarizer** for a memory bank. Your job is to convert one episode into a reusable, general experience.
# Input Data
1. **User Query**: The problem context. 2. **Past Experiences**: Existing rules (indexed as [0], [1]...). 3. **Model Execution**: The reasoning process. 4. **Execution Status**: Success or Failure.
# CRITICAL CONSTRAINTS
1. **NO ANSWER LEAKAGE**: Never mention specific option indices or answer strings.
2. **NO SPECIFICS**: Remove specific numbers/names. Replace with variables/concepts.
3. **NO HALLUCINATION**: Do not invent facts.
# ACTION GUIDELINES

> **### SCENARIO: SUCCESSFUL EXECUTION**
> Extract the underlying **Truth** or **Method**.
> - **Content**: Abstract the logic. If knowledge, extract the core fact.
> - **Constraints**: NO specific options (A/B) or specific numbers/entities.

# MEMORY MANAGEMENT
Compare the new insight with **[Past Experiences]**. Briefly determine whether to **ADD** a new rule or **UPDATE ¡index¿** (replace an existing one).
# Output Format
Strictly follow this structure: `## Analysis ...  ## Experience <experience> <value>...</value> <operation>ADD or UPDATE index</operation> </experience>`

---

# User Prompt
# Task Context
**[User Query]** `{question} {choice_txt}`   **[Past Experiences]** `{memory_content}`
**[Model Execution]** `{trajectory}`

**[Execution Status]**   CORRECT (Success)

# Instruction
First, analyze the execution and compare with Past Experiences. Then, generate the XML block with the experience value and the operation (ADD or UPDATE ¡index¿).

---

**Mem-Optimizer Prompt: Failure Case (Training & Evaluation)**

# System Prompt
# Role
You are an expert **Experience Summarizer** for a memory bank. Your job is to convert one episode into a reusable, general experience.
# Input Data
1. **User Query**: The problem context. 2. **Past Experiences**: Existing rules (indexed as [0], [1]...). 3. **Model Execution**: The reasoning process. 4. **Execution Status**: Success or Failure.
# CRITICAL CONSTRAINTS
1. **NO ANSWER LEAKAGE**: Never mention specific option indices or answer strings.
2. **NO SPECIFICS**: Remove specific numbers/names. Replace with variables/concepts.
3. **NO HALLUCINATION**: Do not invent facts.
# ACTION GUIDELINES

> **### SCENARIO: FAILED EXECUTION**
> Analyze the **Root Cause** of the error.
> - **Content**: Identify the *type* of confusion or trap.
> - **Constraints**: DO NOT simply say "Don't choose X". DO NOT quote the wrong text as a rule.

# MEMORY MANAGEMENT
Compare the new insight with **[Past Experiences]**. Briefly determine whether to **ADD** a new rule or **UPDATE ¡index¿** (replace an existing one).
# Output Format
Strictly follow this structure: `## Analysis ...  ## Experience <experience> <value>...</value> <operation>ADD or UPDATE index</operation> </experience>`

---

# User Prompt
# Task Context
**[User Query]** `{question} {choice_txt}`   **[Past Experiences]** `{memory_content}`
**[Model Execution]** `{trajectory}`

**[Execution Status]**   INCORRECT (Failure)

# Instruction
First, analyze the execution and compare with Past Experiences. Then, generate the XML block with the experience value and the operation (ADD or UPDATE ¡index¿).

*Figure 7.* Comparison of Mem-Optimizer prompt templates for successful (top) and failed (bottom) executions. These templates are employed during both training and evaluation phases to either extract general methodologies or diagnose root causes.

---

**Task & Retrieved Experiences**

**[Task Goal]** Put a clean cloth in countertop.

**[Retrieved Experiences]**

`[0]` *key*: "clean some plate and put it in countertop."
*value*: To clean a plate and place it on a countertop, first locate the plate (e.g., in a cabinet), pick it up, go to the sinkbasin, clean it using the command "clean [plate] with sinkbasin", and **only after cleaning** move it to a countertop. Skipping the cleaning step or placing a dirty plate on the countertop results in failure.

`[1]` *key*: "put a clean knife in countertop."
*value*: To put a clean knife on a countertop, first locate the knife (e.g., on a countertop or in storage), pick it up, go to the sinkbasin, clean it using "clean [knife] with sinkbasin", and **only after cleaning** move it to a countertop. Skipping the cleaning step results in failure.

`[2]` *key*: "clean some pan and put it in countertop."
*value*: To clean a pan and place it on a countertop, first locate the pan, verify its identity by examining it, pick it up, go to the sinkbasin, clean it using "clean [pan] with sinkbasin", and **only after cleaning** move it to a countertop.

---

**UMEM Enhanced Agent: w/ Memory (Success)**

**[Environment]** You are in the middle of a room. You see a cabinet 4, cabinet 3, cabinet 2, cabinet 1, countertop 1, garbagecan 1, handtowelholder 2, handtowelholder 1, sinkbasin 2, sinkbasin 1, toilet 1, toiletpaperhanger 1, and towelholder 1.

**[Execution Trajectory]**

**Step 1:** `go to handtowelholder 1` → You see a handtowel 1.      *Exploring...*

**Step 2:** `take handtowel 1 from handtowelholder 1` → You pick up the handtowel 1.      *Wrong object*

**Step 3:** `go to sinkbasin 2` → You arrive at sinkbasin 1.
**Step 4:** `move handtowel 1 to sinkbasin 1` → You move the handtowel 1 to the sinkbasin 1.
**Step 5:** `take handtowel 1 from sinkbasin 1` → You pick up the handtowel 1.
**Step 6:** `go to countertop 1` → You see a candle 1, a cloth 3, and a spraybottle 1.      *Found cloth!*
**Step 7:** `examine countertop 1` → On the countertop 1, you see a candle 1, a cloth 3, and a spraybottle 1.
**Step 8:** `move handtowel 1 to countertop 1` → You move the handtowel 1 to the countertop 1.

**Step 9:** `take cloth 3 from countertop 1` → You pick up the cloth 3.      *Correct object*

**Step 10:** `go to sinkbasin 1` → You arrive at sinkbasin 1.      *Key Step*

**Step 11:** `clean cloth 3 with sinkbasin 1` → You clean the cloth 3 using the sinkbasin 1.      *Key Step*

**Step 12:** `go to countertop 1` → You arrive at countertop 1.
**Step 13:** `move cloth 3 to countertop 1` → You move the cloth 3 to the countertop 1.

**[Result]** SUCCESS     **Total Steps: 13**     The agent explores, finds the correct object, and applies the cleaning pattern from memory.

---

**Baseline Agent: w/o Memory (Failure)**

**[Environment]** (Same as above)

**[Execution Trajectory]**

**Step 1:** `go to countertop 1` → You see a candle 1, a cloth 3, and a spraybottle 1.
**Step 2:** `take cloth 3 from countertop 1` → You pick up the cloth 3.

**Step 3:** `move cloth 3 to countertop 1` → You move the cloth 3 to the countertop 1.      *Missing clean!*

**Step 4:** `examine countertop 1` → You see a candle 1, a cloth 3, and a spraybottle 1.
**Step 5:** `take cloth 3 from countertop 1` → You pick up the cloth 3.
**Step 6:** `examine cloth 3` → There's nothing special about cloth 3.

**Step 7:** `move cloth 3 to countertop 1` → You move the cloth 3 to the countertop 1.      *Repeated*

**Step 8:** `examine countertop 1` → You see a candle 1, a cloth 3, and a spraybottle 1.

**Step 9:** `help` → Available commands: look, inventory, go to, open, close, take, move, examine, use, heat, clean, cool, slice...      *Ignored clean*

**Step 10:** `take cloth 3 from countertop 1` → You pick up the cloth 3.

**Step 11:** `move cloth 3 to countertop 1` → You move the cloth 3 to the countertop 1.      *Repeated*

**Step 12:** `examine countertop 1` → You see a candle 1, a cloth 3, and a spraybottle 1.
**Step 13:** `inventory` → You are not carrying anything.

**Step 14:** `take candle 1 from countertop 1` → You pick up the candle 1.      *Wrong object*

**Step 15:** `move candle 1 to countertop 1` → You move the candle 1 to the countertop 1.

**...**      *Stuck in take → move loop ×10+*

**Step 30:** `move cloth 3 to countertop 1` → You move the cloth 3 to the countertop 1.

**[Result]** FAILURE     **Total Steps: 30**     The agent never uses the "clean" command despite seeing it in help.

---

*Figure 8.* Case study comparing UMEM enhanced agent and baseline.

