# OpenReview forum: "UMEM: Unified Memory Extraction and Management Framework for Generalizable Memory"
_ICML.cc/2026/Conference — ICML 2026 regular_

### Official Review · Reviewer_6T2B · 2026-02-27

**Soundness:** 2
**Presentation:** 2
**Significance:** 2
**Originality:** 2
**Overall Recommendation:** 2
**Confidence:** 4

**Summary:**

This paper proposes the UMEM (Unified Memory Extraction and Management) framework to address the joint optimization of memory extraction and memory management in self-evolving agents. The authors argue that most existing approaches treat memory extraction as a static process (invoking off-the-shelf LLMs via prompts) and focus only on optimizing memory management strategies. As a result, agents tend to accumulate instance-specific noise rather than generalizable knowledge, which undermines their ability to generalize across tasks.

**Compliance With Llm Reviewing Policy:**

Affirmed.

**Final Justification:**

There are unresolved issues with the experimental data.

**Key Questions For Authors:**

1. Some experimental results (e.g., certain metrics marked with ↓ in Table 1) are not sufficiently discussed to explain why performance degrades.
2. The pseudocode in Algorithm 1 is slightly inconsistent with the main text (e.g., the order of reward computation).
3. Memories are extracted from trajectories using the Mem-Optimizer. However, since the Mem-Optimizer has relatively weak language capabilities, it may generate ambiguous or incorrect content and may even output repeated tokens. How is this issue addressed?
4. During agent execution, the Mem-Optimizer is continuously updated. Repeatedly optimizing Equation (1) with GRPO could cause catastrophic forgetting, gradually degrading the model’s core language ability. Ultimately, the Mem-Optimizer’s memory-extraction quality may deteriorate over time, or even collapse entirely.

**Limitations:**

1. The memory mechanism may be exploited maliciously: if an attacker “injects” incorrect memories through a crafted sequence of queries, it could influence the agent’s subsequent decisions. It would be helpful to discuss memory verification or adversarial robustness.
2. Semantic neighborhood modeling relies on a pretrained encoder; if the encoder exhibits biases (e.g., cultural or linguistic bias), these may be amplified in the memory system.
3. Does the “monotonic improvement” assumption of the online evolution mechanism still hold in out-of-distribution settings? Consider discussing potential degradation modes under distribution shift.

**Strengths And Weaknesses:**

## Strengths:
- The problem is clearly defined. The paper draws an analogy between self-evolving agents and the forward inference + backward optimization process in neural networks, resulting in a sound theoretical framing.
- The experimental design is fairly comprehensive: 5 benchmarks, 3 executor models, and coverage of both multi-round and single-round tasks.
- The ablation studies are well-designed, validating the contribution of each component.

## Weaknesses:
1. The design of the efficiency regularization term in the marginal-utility reward feels somewhat heuristic, and the paper lacks a theoretical justification for why reducing tokens should represent "efficient reasoning."
2. Is the second formula in the **Task Formulation of Self-Evolving Agents** section incorrect? It seems to be missing the retrieved memories.

---

> ### Author Rebuttal · Authors · 2026-03-31
>
> **Q1:** Why does reducing tokens represent more efficient reasoning?
>
> **A1:** Using reduced token count as a metric for reasoning efficiency is established practice [1]. In UMEM, we reward memories that minimize token usage without compromising accuracy, effectively helping the executor bypass redundant planning or ineffective paths. Our ablation confirms this: removing the efficiency reward on Qwen3-8B-Thinking caused accuracy to drop by **1.70%** while increasing token consumption by **509.60**. This demonstrates that penalized tokens correspond to unnecessary reasoning, and this term successfully reduces execution overhead (**see A3 to Reviewer qACS**).
>
> [1] Efficient Reasoning Models: A Survey.
>
>
> **Q2:** On the occasional instability in some settings.
>
> **A2:**
> We attribute the occasional instability mainly to the limited capacity of the 1B Mem-Optimizer. We verify this by scaling the memory model from 1B to 14B: under the same setup, **UMEM consistently outperforms No-Train-14B on all benchmarks**, and the previous fluctuations disappear. The detailed results are provided in **A3 to Reviewer LQoN**.
>
> **Q3:** Clarification on minor typos.
>
> **A3:**
> We appreciate the feedback. We will correct the Section 3 formula to include the retrieved memory set $\mathcal{B}_t^{topk}$ in the executor input. We will also clarify that while the text introduces Marginal Utility Reward (MUR) first for readability, Algorithm 1 and our implementation prioritize the format reward.
>
> **Q4:** Memory generation quality of the small model
>
> **A4:**
> We agree that small models can produce lower-quality memories, but in UMEM these errors are actively corrected during training rather than simply accumulated. First, under GRPO, low-quality memories receive low marginal utility reward because they fail to provide valuable feedback to semnatic neighborhood, so the policy is directly pushed away from such generations. Second, UMEM’s online UPDATE mechanism can revise previously stored low-quality memories within the current Top-K retrieved set, instead of leaving them in the bank unchanged. This is exactly the motivation of neighborhood-level reward and joint extraction-management optimization in UMEM.
>
> We also tracked several training diagnostics, which collectively prove that the Mem-Optimizer is successfully optimized to generate more fluent and reuseable memory content for future tasks: PPL decreases from 10.02 to 8.56, format reward rises from 0.81 to 1.00 (reaching 0.95 by step 2), 10-gram repetition stays low (avg. 0.0097), and execution accuracy over semantic neighborhoods improves by about 9%. **These results suggest that low-quality memories are effectively corrected during UMEM training**, even with a small model.
>
> We also provide Fig. 1 at:
> https://anonymous.4open.science/r/ICML_2026_Rebuttal-D242/training_dynamic.png
>
>
>
> **Q5:** Does UMEM suffer from catastrophic forgetting?
>
> **A5:**
> We do not observe evidence that UMEM induces catastrophic forgetting. As shown in Fig. 1, execution accuracy over semantic neighborhoods increases by ∼9% during training, indicating that memory quality improves rather than degrades.
> Moreover, our main evaluations are on OOD tasks; if the learned memories had deteriorated, such gains would be unlikely to persist.
>
> In addition, UMEM uses RL-based optimization, which updates only a small fraction of parameters (5.89% above the 1e−5 threshold) and is generally less prone to catastrophic forgetting than SFT [2-3]. Finally, to assess potential catastrophic forgetting, we evaluate on IFEval and HumanEval, where UMEM-Qwen3-4B improves over Qwen3-4B.
> We will incorporate these discussions and findings into the camera-ready version.
>
> |Model|IFEval(Strict Avg)|HumanEval(Pass@1)|
> |-|-|-|
> |Qwen3-4B|85.53%|63.53%|
> |**UMEM-4B**|**87.33%**|**65.85%**|
>
> [2] RL's Razor: Why Online Reinforcement Learning Forgets Less.
>
> [3] Reinforcement Learning Finetunes Small Subnetworks in Large Language Models.
>
>
> **Q6:** Additional discussion on cross-task generalization
>
> **A6:**
> To quantify generalization, we evaluate **Cross-Task Utility** (accuracy using memories from different prior benchmarks) and **Error Recovery Rate** (percentage of no-memory failures resolved by transferred memory). Averaged over randomized sequences, UMEM-4B achieves 41.47% accuracy (vs. 39.29% No-Train) and a 23.17% recovery rate (vs. 17.80% No-Train), please **refer to A1.3 to Reviewer LQoN**.
>
> Notably, UMEM-4B's cross-benchmark accuracy slightly exceeds its single-benchmark performance, while the No-Train baseline degrades. This confirms UMEM extracts **reusable** memory patterns rather than task-specific overfits. However, transfer utility is not strictly monotonic, as it is inherently constrained by inter-benchmark similarity. **See A5 to Reviewer qACS** for detailed discussions on impact and limitations.
>
> |Method|single-bench avg|cross-bench avg|
> |-|-|-|
> |**UMEM-4B**|**57.74**|**58.33**|
> |No-Train|53.18|52.92|
> |No-Memory|45.36|45.36|

---

> > ### Author Rebuttal · Reviewer_6T2B · 2026-04-03
> >
> > There are unresolved issues with the experimental data.

---

### Official Review · Reviewer_qACS · 2026-02-28

**Soundness:** 2
**Presentation:** 3
**Significance:** 3
**Originality:** 2
**Overall Recommendation:** 4
**Confidence:** 2

**Summary:**

This paper presents UMEM, a self-evolving LLM agent framework. It jointly optimizes memory extraction and management via a learnable Mem-Optimizer trained with GRPO. To improve generalization and reduce overfitting, candidate memory updates are evaluated across a semantic neighborhood of related queries, with a marginal utility reward that integrates success correction and efficiency regularization. Experiments on five benchmarks (single-turn reasoning, multi-turn embodied tasks) and three executors verify consistent improvements over strong baselines. Ablation studies validate the effectiveness of joint optimization and neighborhood-based rewards.

**Compliance With Llm Reviewing Policy:**

Affirmed.

**Final Justification:**

The authors have addressed my concerns. While I still share the same concerns as Reviewer 6T2B, I will maintain my positive assessment of the paper.

**Key Questions For Authors:**

1. How are memory keys constructed for newly inserted memory entries, and what specific retrieval index is employed over the memory bank during inference (including the choice of encoder, key representation, and how keys are updated upon the UPDATE operation)?

2. Could you present an ablation study on the efficiency regularization term \(R_{\text{eff}}\), such as results with different weighting coefficients or complete removal, as well as its impact on accuracy across each benchmark?

3. What is the detailed breakdown of computational cost (e.g., executor calls per training step: \(G\) rollouts × \(N\) neighbors × 2 states, plus basic rollouts)? How does the overall training time scale with parameters \(N\), \(G\), and \(K\)?

4. Were all baseline methods evaluated using the same streaming protocol with online memory updates and identical retrieval encoders? For fair comparison, please provide their hyperparameters, memory budgets, and prompt designs.

**Limitations:**

The author did not fully discuss the limitations and potential negative social impacts of their work. Please add them in the revised version.

**Strengths And Weaknesses:**

### Strengths
1. Compared with existing methods that only optimize memory selection/management while keeping extraction fixed, the joint learning of memory extraction and management under a unified policy is an innovative and up-to-date research direction.
2. Extensive experiments on diverse tasks (mathematics, science, multi-hop QA, embodied ALFWorld) and different executors demonstrate the framework’s good portability and model-agnostic advantages.
3. The overall pipeline, training process, and reward design are clearly described, and the figures effectively illustrate the motivation and logic of neighborhood validation.

### Weaknesses
1. The details of memory key construction and retrieval indexing are insufficiently clarified, such as how keys are generated for new memory entries, whether keys are derived from queries or learned summaries, and how retrieval works on mixed content.
2. The fairness and experimental settings of baseline methods (e.g., ReMem, MemP) are not fully explained, especially regarding prompts, retrieval backends, and whether baselines can also use streaming memory updates under the same conditions.
3. There are still some repeated sentences and minor formatting issues; more precise definitions of the memory schema (including key/value fields and schema evolution) are needed to reduce ambiguity.

---

> ### Author Rebuttal · Authors · 2026-03-31
>
> **Q1:** How are memory keys constructed and how does retrieval over the memory bank work?
>
> **A1:**
> Memory schema and operations follow the best practice of research community [1], Specifically:
>
> 1. Memory Format: UMEM employs a key-value bank where keys are query embeddings (BGE-M3) and values are extracted memory contents.
>
> 2. Retrieval: For each input query, the Top-K most relevant key-value pairs are retrieved and provided to the executor as context.
>
> 3. Memory Operations: UMEM jointly optimizes extraction and management through execution-aware operations:
>
>     - ADD: If Top-K memories are insufficient for successful execution, a new memory is created from the trajectory and inserted.
>
>     - UPDATE: If a retrieved entry is relevant but suboptimal, UMEM revises its value while keeping the original key fixed.
>
> This local, execution-aware optimization ensures UMEM only modifies memories directly impacting the task, avoiding global interference and improving memory reusability. We will supplement these details in our revision.
>
> [1] Memp: Exploring Agent Procedural Memory
>
> **Q2:** Baseline configuration details.
>
> **A2:**
> To ensure a fair comparison, we employ a unified retriever (BGE-M3), identical execution prompts, and 0.0 decoding temperature. For all baselines, we strictly follow original paper settings with an unlimited memory budget. Notably, all selected methods natively support our streaming protocol (sequential updates and retrieval), ensuring a consistent evaluation across all benchmarks. These details will be fully clarified in the camera-ready version.
>
>
> **Q3:** Ablation on removing the Efficiency Regularization.
>
> **A3:**
> Averaged across all five benchmarks, removing the efficiency reward degrades performance for both models: GPT-5.1 accuracy **drops by 1.60% (+29.28 tokens)** and Qwen3-8B-Thinking **by 1.70% (+509.60 tokens)**. We hypothesize that high-quality memory helps the model "prune" redundant reasoning paths, whereas low-quality context introduces noise that triggers longer, less focused reasoning traces. These results will be included in the camera-ready version.
>
> |Method|GPT-5.1 (Acc. Avg)|GPT-5.1 (Token Avg)|Qwen3-8B-Thinking (Acc.Avg)|Qwen3-8B-Thinking (Token Avg)|
> |-|-|-|-|-|
> |**UMEM-Qwen-4B**|**57.74**|**1,473.40**|**51.13**|**7,590.00**|
> |w/o Length Reward|56.14|1,502.68|49.43|8,099.60|
>
>
> **Q4:** Computational Cost Breakdown and Scaling Analysis
>
> **A4:**
> A detailed breakdown of our training subtasks is provided in **A1 to Reviewer d1xT** (UMEM speed: **15.83 min/step**). Per-step executor calls total E + 2\*G\*N: E=1 initial rollout for memory extraction, plus 2\*G\*N calls evaluating G GRPO samples on N semantic neighbors under both with- and without-memory conditions.
>
> The following tables detail our scaling analysis: increasing N or G linearly boosts reward-evaluation and GRPO optimization costs. Increasing K only slightly extends prompt contexts with minimal runtime impact, as it avoids additional executor calls.
>
> **Subtask runtime breakdown (N & G)**
>
> *Fixed E=1, G=8, scale N. Avg time (s) per subtask.*
>
> |Subtask|N=1|N=3|N=5|
> |-|-|-|-|
> |Executor rollout|182.70|175.90|171.00|
> |Mem-Optimizer rollout|26.20|27.50|29.80|
> |Reward evaluation|316.00|623.90|970.80|
> |GRPO optimization|101.80|108.00|116.20|
> |Memory update|0.50|0.80|0.70|
> |Other overhead|0.00|13.50|0.00|
> |Total|627.30|**949.60 (ours)**|1,288.40|
>
>
> *Fixed N=3, E=1, scale G. Avg time (s) per subtask.*
>
> |Subtask|G=4|G=8|G=16|
> |-|-|-|-|
> |Executor rollout|199.30|175.90|183.00|
> |Mem-Optimizer rollout|22.10|27.50|40.70|
> |Reward evaluation|364.10|623.90|1,434.10|
> |GRPO optimization|52.60|108.00|227.20|
> |Memory update|0.60|0.80|0.70|
> |Other overhead|0.00|13.50|0.00|
> |Total|638.70|**949.60 (ours)**|1,885.60|
>
> **Q5:** Discussion on limitations and social impact.
>
> **A5:**
> We agree that discussing limitations and social impact is essential and will include the following section in our camera-ready version:
>
> **Limitation & Impact Statement.**
>
> While UMEM remains effective across different benchmarks, the level of improvement depends on the semantic similarity between queries, as a memory must first be retrieved before its generalizability can take effect. Future research could explore more effective retrieval methods to expand the utility range of generalized memories and mitigate potential biases (e.g., cultural or linguistic) from pretrained encoders. Additionally, this mechanism faces risks of malicious exploitation, where an attacker might "inject" incorrect memories through crafted query sequences to mislead an agent's decisions. Incorporating memory verification or adversarial robustness training would be a valuable direction to ensure the integrity of the memory-augmented process.

---

> > ### Author Rebuttal · Reviewer_qACS · 2026-04-03
> >
> > The authors have addressed my concerns. However, based on the overall innovation and feedback of the paper, I tend to maintain the original score.

---

### Official Review · Reviewer_LQoN · 2026-03-12

**Soundness:** 2
**Presentation:** 3
**Significance:** 3
**Originality:** 3
**Overall Recommendation:** 3
**Confidence:** 3

**Summary:**

The paper identifies that treating memory extraction as a static process in self-evolving LLM agents introduces instance-specific noise. To address this, the authors propose UMEM, a framework that employs a trainable Mem-Optimizer to jointly optimize memory extraction and management using GRPO. By evaluating memory updates based on their utility across semantic neighborhoods rather than single instances, UMEM cultivates more generalizable memory banks and surpasses baselines such as ReMem and Memp across diverse benchmarks and models.

**Compliance With Llm Reviewing Policy:**

Affirmed.

**Key Questions For Authors:**

Q1: Could the authors explain why a reward based on semantic neighborhoods effectively encourages the extraction of truly generalizable knowledge, rather than simply ensuring that similar queries retrieve similar memories?

Q2: Why were closely related and contemporary methods cited in the paper, such as ReasoningBank, EvolveR, Memory-R1, and especially MemRL, not included in the experimental comparisons?

Q3: What factors cause UMEM to underperform compared to the No Memory baseline in specific scenarios (e.g., with Qwen3-8B on ALFWorld and GPQA), and under what conditions does the method typically fail?

Q4: Given that continual evolution is highly sensitive to initial states and sample ordering, could the authors provide statistical evidence (e.g., means, variances, or significance tests) to confirm the stability of their findings?

Q5: How can the performance gains be attributed to the UMEM framework itself rather than the specific skill-level prompt design used for the Mem-Optimizer, which explicitly instructs the model to be more general?

Q6: To support the claim of superior memory quality, could the authors provide a direct comparison of the memory content extracted by UMEM versus other methods under identical conditions to demonstrate its higher level of abstraction?

**Limitations:**

No. The authors should discuss the limitations of UMEM regarding cross-task generalization. Specifically, evaluating whether the general memories extracted by the Mem-Optimizer remain effective under task distribution shifts would provide stronger evidence of their true generality.

**Strengths And Weaknesses:**

### Strengths

1. The paper addresses a bottleneck in self-evolving agents by shifting the focus from simple memory management to a joint optimization perspective, tackling the lack of generalization in traditional static memory extraction.

2. The design of UMEM is logically sound, particularly through its semantic neighborhood modeling and the unified approach of generating both memory content and management operations within a single framework.

3. The study provides a robust evaluation by covering both single-turn reasoning and multi-turn interaction scenarios, supplemented by a long-term evolution analysis that underscores the agent's sustained performance.

### Weaknesses

1. While the semantic neighborhood modeling is intuitive, the theoretical argument primarily shows that similar queries retrieve similar memories. This does not prove that defining rewards this way guarantees the extraction of more generalizable or useful knowledge for future tasks.

2. The evaluation omits several highly relevant and contemporary methods cited in the paper (such as ReasoningBank, EvolveR, Memory-R1, and MemRL), leaving the claims of UMEM’s superiority over the current state-of-the-art poorly supported.

3. In many benchmarks, the absolute improvement of UMEM over existing baselines and the No Memory baseline is relatively small, suggesting that the method may not offer a stable or significant practical advantage in its current form.

4. UMEM occasionally underperforms compared to the No Memory baseline in specific configurations (e.g., with Qwen3-8B-Thinking on ALFWorld and GPQA), indicating that the method's benefits are not universal across different tasks and models.

5. The paper fails to report means, variances, or significance tests over multiple runs, which is a critical omission for continual evolution experiments that are highly sensitive to random seeds, sample ordering, and initial memory states.

6. It remains unclear whether the observed gains result from the proposed joint optimization framework or simply from the sophisticated skill-level extraction prompts used for the Mem-Optimizer, necessitating more rigorous ablation of the prompt design itself.

7. The existing case studies only compare the agent's performance with and without memory. They do not directly compare the content of memories extracted by UMEM against those of other methods to demonstrate superior extraction quality or higher levels of abstraction.

---

> ### Author Rebuttal · Authors · 2026-03-31
>
> **Q1:** Why does Semantic Neighborhood Modeling (SNM) encourage generalizable memory?
>
> **A1:**
>
> 1. **Modeling Future Reuse via SNM**: In retrieval-based systems, a memory entry’s utility is realized when it is retrieved and reused by future queries. UMEM’s SNM explicitly models this by using a Cosine Neighborhood as a proxy for the future query distribution. As proven in Appendix C, this ensures that memory optimizations are calibrated for the specific set of semantically related queries that will actually rely on them.
>
> 2. **Neighborhood-Level Reward Optimization**: UMEM maximizes a memory's average utility across its semantic neighborhood. Under GRPO, instance-specific noise that overfits the source query fails to improve this collective reward and is naturally penalized. This forces the Mem-Optimizer to discard task-specific trivia and distill genuinely reusable patterns that provide stable benefits across all related queries.
>
> 3. **Empirical Evidence of Cross-Task Generalization**:
> Empirically, UMEM significantly improves out-of-distribution (OOD) performance across five benchmarks and unseen executors (Table 1), while ablating SNM causes severe degradation (Table 2). To further quantify generalization, we evaluate UMEM across two key metrics:
>
>    - **Cross-Task Utility** (CTU): Accuracy on OOD tasks using memories from entirely different prior benchmarks (e.g., testing on Math tasks using memories previously generated during Physics tasks).
>
>    - **Error Recovery Rate** (ERR): The percentage of no-memory failures resolved by transferred memory (i.e., cases where the model fails without memory on the current task but succeeds using knowledge transferred from other tasks).
>
> Averaged over three randomized sequences to eliminate order bias, UMEM demonstrates superior OOD generalization:
>
> |Method|CTU(%)|ERR(%)|
> |-|-|-|
> |**UMEM-4B**|**41.47**|**23.17**|
> |No-Train-4B|39.29|17.80|
>
>
>
> **Q2:** Baseline selection and result stability.
>
> **A2:**
> We excluded EvolveR as it relies on search engines and online-trainable executors, which are incompatible with our frozen LLM and zero-start setting; however, we now include its memory strategy for comparison. ReasoningBank and MemRL were omitted due to post-submission code releases.
> Averaged over three randomized runs, UMEM demonstrates robust OOD generalization. Statistical analysis confirms significant gains over top baselines: ReasoningBank for Gemini-2.5-Flash ($p < 0.05$) and EvolveR for GPT-5.1 ($p < 0.02$).
>
> |Method|GPT-5.1|Gemini-2.5-Flash|
> |-|-|-|
> |No Memory|45.36|49.14|
> |ReasoningBank|53.76 ± 0.82|54.39 ± 0.59|
> |MemRL|45.91 ± 0.59|51.67 ± 0.87|
> |EvolveR|55.34 ± 0.58|52.42 ± 0.47|
> |Memp|54.24 ± 0.73|51.60 ± 0.58|
> |Qwen3-4B-Instruct|53.67 ± 0.49|51.00 ± 0.91|
> |UMEM-Qwen3-4B (Ours)|**58.10** ± 0.72|**55.46** ± 0.82|
>
>
> **Q3:** What causes UMEM's occasional underperformance or failure?
>
> **A3:**
> The occasional underperformance in Table 1 is primarily due to the limited capacity of 1B models, which lack the parameter scale to robustly extract generalizable patterns.
>
> We verified this by replacing the 1B Mem-Optimizer with a 14B model (using the same Qwen3-8B-Thinking executor), which eliminated the degradation and yielded stable gains. These results confirm that UMEM’s performance scales positively with the Mem-Optimizer’s size. We will include this scaling analysis and stability discussion in our revision.
>
> |Model|Stat|AIME|GPQA-Diamond|HLE|HotpotQA|ALFWorld SR|ALFWorld PR|
> |-|-|-|-|-|-|-|-|
> |Qwen3-14B|Mean|57.22|53.71|7.84|61.67|47.51|70.69|
> ||std_err|0.45|0.14|0.14|0.27|0.89|0.80|
> |UMEM-Qwen3-14B|Mean|**60.56**|**55.05**|**8.74**|**63.67**|**52.24**|**74.55**|
> ||std_err|0.45|0.29|0.52|0.27|0.93|0.42|
> |Δ|Mean|+3.34|+1.34|+0.90|+2.00|+4.73|+3.86|
>
>
> **Q4:** Does UMEM's gain stem from prompt design or the optimization framework?
>
> **A4:**
> UMEM outperforms the No-Train baseline (Line 264), which uses the same prompt but without training. This confirms that gains are driven by the Joint Optimization Framework, not merely prompt engineering.
>
> **Q5:** More case studies.
>
> **A5:**
> Case analysis confirms UMEM extracts universal patterns where baselines overfit.
>
> > In a GPQA physics query, No-train retained source-specific formula residue ($m_{\nu}^2$ corrections) from a neutrino scattering task, leading to a crude 2 MeV estimate. In contrast, **UMEM extracted the core reasoning by transitioning from approximations to full relativistic kinematics**, guiding the executor to correctly apply decay-at-rest physics for the precise 5 MeV result.
>
> Due to rebuttal space limits, we will supplement the following in our revision:
>
>    - Cross-Task Utility: Analyzing how UMEM assists downstream execution versus how poorly generalized baseline memories introduce interference or noise.
>
>    - Abstraction Comparison: A head-to-head comparison on identical problems, demonstrating UMEM's superior abstraction level and its impact on enhanced generalization.

---

> > ### Author Rebuttal · Reviewer_LQoN · 2026-04-04
> >
> > My concerns remain in the following parts:
> > 1. The authors omitted the requested comparison with Memory-R1 without providing any justification.
> > 2. The failure analysis relies on scaling up model parameters instead of explaining the actual algorithmic vulnerabilities, and pairing a massive 14B memory-optimizer with a smaller 8B base executor is impractical and counterintuitive.
> > 3. The case study fails to prove superior abstraction, and it only compares extracted memory against an untrained baseline rather than competitive methods.
> > 4. The new generalization metrics (CTU and ERR) lack comparisons against strong baselines, testing UMEM only against the untrained baseline.

---

> > > ### Author Response · Authors · 2026-04-07
> > >
> > > **Q1:** Comparison with Memory-R1?
> > >
> > > **A1:**
> > > We omitted Memory-R1 because its **source code is unavailable**. An unofficial re-implementation would lack strict variable control, making it impossible to separate true algorithmic flaws from implementation discrepancies. Furthermore, Memory-R1 is evaluated entirely on dialogue-centric benchmarks, creating a fundamental domain mismatch with our tasks. Thus, we prioritize verifiable open-source baselines to ensure fair and reproducible evaluation.
> > >
> > > **Q2:** Failure Analysis
> > >
> > > **A2:**
> > > While scaling models helps improve overall stability, we agree that analyzing specific failures is essential. To understand UMEM's limitations, we focus on **two specific error types**: cases where UMEM fails but either the NoMemory or NoTrain baseline succeeds. Though infrequent (accounting for ~7% of cases in UMEM-1B and ~5% in UMEM-4B), these errors reveal two key issues:
> > >
> > >    **1. Memory Interference (NoMemory correct, UMEM fails):** When the prompt already contains all necessary information (e.g., extractive QA or self-contained reasoning), injecting external memory can distract the executor. This suggests memory usage should not be forced by default. A natural solution is to encapsulate memory retrieval as a callable tool, allowing the base model to autonomously decide whether to invoke it based on the task.
> > >
> > >    **2. Abstraction vs. Precision Trade-off (NoTrain correct, UMEM fails):**
> > >    By filtering surface details, UMEM intentionally prioritizes broad generalization over rote memorization. As a result, NoTrain occasionally outperforms it on specific queries requiring rigid pattern matching (e.g., isolated ALFWorld instances) because it preserves unabstracted raw data. Furthermore, extracting raw experiences into generalizable rules is inherently challenging; consequently, a smaller memory extractor like UMEM-1B struggles more with this delicate process, introducing slightly more extraction imperfections.
> > >
> > > Ultimately, both failure modes highlight that memory should be a selective aid rather than a default mechanism. The core challenge lies in dynamically determining not only whether to invoke memory, but also the appropriate level of abstraction for a given task. We will explicitly discuss these limitations in a dedicated Section 5.7 (Failure Analysis) in the final version.
> > >
> > > **Q3:** On Superior Abstraction and Baseline Comparisons.
> > >
> > > **A3:**
> > > Our case study highlights a challenging cross-task transfer scenario: applying a memory extracted from a specific HLE problem (neutrino-scattering approximation) to an unseen OOD GPQA problem (spontaneous-fission kinematics). Instead of finding the optimal abstraction, baselines polarize into two extremes:
> > >
> > > **1. Too Task-Specific:** These methods overfit to the source query by memorizing surface details or rigid procedural checklists.
> > >    * No-train copies source-formula residue (e.g., $m_{\nu}^2$ correction terms) that does not transfer to fission.
> > >    * MemRL stores a narrow error log instead of a reusable strategy.
> > >    * MemP generates a rigid checklist that only works for the original formula.
> > >    * ReasoningBank breaks the experience into surface-level math rules rather than extracting the core physical principles.
> > >
> > > **2. Too Generic:** Over-compression strips away actionable guidance, leaving the executor without concrete logical steps.
> > >    * EvolveR strips away the math, leaving only broad warnings (e.g., "avoid pattern matching") that cannot guide execution.
> > >
> > > **3. The Optimal Abstraction:** UMEM hits the sweet spot by extracting the underlying physical rule rather than surface-level math. It learns that when an approximation fails, one must switch to exact energy-momentum conservation. This rule tells the agent exactly what to do, yet it is broad enough to transfer across different physical scenarios, successfully solving the unseen GPQA target and recovering the correct $5\,\mathrm{MeV}$ answer.
> > >
> > > **Q4:** Comparison of generalization metrics (CTU and ERR) against strong baselines.
> > >
> > > **A4:**
> > > The results distinguish "doing no harm" (CTU) from "providing value" (ERR). Strong baselines like MemRL maintain high CTU through reranking mechanisms, which avoids accuracy drops by discarding irrelevant memories. However, "avoiding harm" does not equate to "fixing errors," as reflected in their stagnant ERR.
> > >
> > > In contrast, UMEM achieves a 5.87% to 11.57% ERR lead over all baselines. By optimizing for neighborhood-level utility, UMEM extracts transferable patterns that actively **resolves** failures in OOD tasks, rather than merely bypassing them. This significantly enhances memory reusability, allowing UMEM to fix errors in new domains where baselines' task-specific memories **fall short**. We will include this detailed discussion in the camera-ready version.
> > >
> > > |Method|CTU(%)|ERR(%)|
> > > |-|-|-|
> > > |**UMEM-4B(Ours)**|**41.47**|**23.17**|
> > > |MemRL|40.40|17.30|
> > > |No-Train-4B|39.29|17.80|
> > > |MemP|37.20|16.00|
> > > |EvolveR|30.30|15.10|
> > > |ReasoningBank|18.20|11.60|

---

### Official Review · Reviewer_d1xT · 2026-03-13

**Soundness:** 4
**Presentation:** 4
**Significance:** 3
**Originality:** 3
**Overall Recommendation:** 5
**Confidence:** 4

**Summary:**

This paper introduces  UMEM (Unified Memory Extraction and Management). UMEM is a self-evolving agent framework designed to improve how large language model (LLM)–based agents learn from experience through external memory (key-value pairs). Its core idea is to optimizes both memory extraction and management within a single learned Mem-Optimizer, while keeping the task executor frozen.

During training, the authors propose (i) the construction of a Semantic Neighborhood for evaluating more than one queries (and the memory updates these would incur)  - which are semantically close to the input one, (ii) a Marginal Utility Reward optimized via GRPO and (iii) continuous updates of the memory bank. By integrating these components in joint memory extraction and management training, the memory agent generalizes better. This is quantified in extensive benchmarks and against SOTA baselines (up to a 10.67% improvement in multi-turn interactive tasks).

**Compliance With Llm Reviewing Policy:**

Affirmed.

**Final Justification:**

I maintain my positive score, guided by my overall assessment in the original reviews and the authors’ subsequent reactions.

**Key Questions For Authors:**

## Questions
1. It would be highly informative for the reader to include a short discussion on the training cost incurred in UMEM (ideally itemized for the additional "subtasks") as compared to that of comparison baselines.

**Strengths And Weaknesses:**

## Strengths

- Using Semantic Neighborhood, memory updates are evaluated over clusters of queries that are semantically similar and then their marginal utility reward is optimized using GRPO: this approach critically mitigates instance specific noise.
- Joint optimization of memory extraction and update (as in the "unified" Mem-Optimizer) is a novel feature: it target misalignment of extracted content and downstream memory operations.
- Empirical validation is strongly favorable for UMEM for single-turn tasks and multi-turn embodied settings over strong baselines (including SOTA memory management systems like ReMem and Memp). Additionally, ablation studies cleanly quantify the role of Semantic Neighborhood and detail the relative importance of memory extraction and memory management in joint optimization, by reporting the performance degradation when these components are removed.


## Weaknesses

- Training is computationally demanding (semantic neighborhood, rollouts, GRPO optimization, memory updates), however no related discussion is included in the main paper.
- It is not clear how sensitive UMEM is to changes in computing Semantic Neighborhood as a result for example of changes in the embeddings encoder or the similarity metric.

---

> ### Author Rebuttal · Authors · 2026-03-31
>
> **Q1:** Itemized Breakdown of Training Cost
>
> **A1:**
> We agree that UMEM introduces additional computation and will include the following per-step wall-clock breakdown in our camera-ready version. Profiled on 8× A100 (80GB) GPUs using the anchor setting ($N=3$ semantic neighbors, $E=1$ executor rollout, $G=8$ GRPO rollouts) described in Appendix A, the primary bottleneck is Reward Evaluation (65.7%). This is primarily due to the $G$ GRPO rollouts and $G \times N$ total executor executions required to compute neighborhood-level rewards. We view this overhead as a necessary trade-off for the observed performance gains. For a detailed analysis of how **scaling $G$ and $N$** impacts training costs, please **refer to A4 to Reviewer qACS**.
>
> **UMEM per-step runtime breakdown**
>
> | Subtask | Avg time (s / step) | Share of total |
> | :--- | :---: | :---: |
> | Initial executor rollout | 175.90 | 18.5% |
> | Mem-Optimizer rollout | 27.50 | 2.9% |
> | Reward evaluation | 623.90 | 65.7% |
> | GRPO optimization | 108.00 | 11.4% |
> | Memory update | 0.80 | 0.1% |
> | Other overhead | 13.50 | 1.4% |
> | Total | 949.60 | 100.0% |
>
>
> **Q2:** Sensitivity to embedding encoder choices.
>
> **A2:**
> To evaluate sensitivity to the embedding encoder, we conducted an ablation study replacing the default BGE-M3 with Qwen3-Embedding-0.6B (which has a similar parameter size). We measured the optimized Mem-Optimizer's average downstream performance across 5 benchmark metrics, using GPT-5.1 and Qwen3-8B-Thinking as frozen executors. As shown below, performance remains stable and highly comparable, demonstrating that **UMEM is not overly dependent on a specific encoder**. These results will be incorporated into Table 2 in our camera-ready version.
>
> | Method | GPT-5.1 (Avg) | Qwen3-8B-Thinking (Avg) |
> | :--- | :--- | :--- |
> | UMEM-BGE-M3 | 57.74 | 51.13 |
> | UMEM-Qwen3-Embedding-0.6B | 58.36 | 51.10 |

---

> > ### Author Rebuttal · Reviewer_d1xT · 2026-04-04
> >
> > The authors have addressed my concerns, and I will therefore maintain my positive original score.

---

### Decision · Program_Chairs · 2026-04-30

**Decision:**

Accept (regular)

**Comment:**

The majority of reviewers find the work technically solid and significant. The joint optimization framework to train a model for memory management is a step forward for agentic memory. In particular:

* Methodological Novelty: Shifting from heuristic memory management to a learned, unified policy is a timely  contribution.

* Empirical Breadth: Evaluated across 5 benchmarks and 3 executors, showing marginal but consistent gains agains alternative memory management approaches. .

* Strong Rebuttal: The authors provided extensive new data (scaling the model, OOD metrics CTU/ERR, and training diagnostics) that addressed most concerns regarding cost and generalization.


Concerns / Points of discussion:

* Baseline Comparisons: While some contemporary baselines were missing in the initial draft, the authors added comparisons to ReasoningBank and EvolveR during the rebuttal, strengthening the significance.

* While conceptual concerns were generally resolved, Reviewer 6T2B raised some additional concerns regarding reproducability/baseline comparisons that could not be resolved during the rebuttal.